# Cancer-derived small extracellular vesicles promote angiogenesis by heparin-bound, bevacizumab-insensitive VEGF, independent of vesicle uptake

Song Yi Ko [1], WonJae Lee[1], Hilary A. Kenny[2], Long H. Dang[3], Lee M. Ellis[1,4], Eric Jonasch[5], Ernst Lengyel[2] & Honami Naora [1]*

Cancer-derived small extracellular vesicles (sEVs) induce stromal cells to become permissive for tumor growth. However, it is unclear whether this induction solely occurs through transfer of vesicular cargo into recipient cells. Here we show that cancer-derived sEVs can stimulate endothelial cell migration and tube formation independently of uptake. These responses were mediated by the 189 amino acid isoform of vascular endothelial growth factor (VEGF) on the surface of sEVs. Unlike other common VEGF isoforms, $VEGF_{189}$ preferentially localized to sEVs through its high affinity for heparin. Interaction of $VEGF_{189}$ with the surface of sEVs profoundly increased ligand half-life and reduced its recognition by the therapeutic VEGF antibody bevacizumab. sEV-associated VEGF (sEV-VEGF) stimulated tumor xenograft growth but was not neutralized by bevacizumab. Furthermore, high levels of sEV-VEGF were associated with disease progression in bevacizumab-treated cancer patients, raising the possibility that resistance to bevacizumab might stem in part from elevated levels of sEV-VEGF.

[1] Department of Molecular and Cellular Oncology, University of Texas MD Anderson Cancer Center, Houston, Texas, USA. [2] Section of Gynecologic Oncology, Department of Obstetrics and Gynecology, University of Chicago, Chicago, Illinois, USA. [3] Division of Hematology and Oncology, Department of Internal Medicine, University of Florida, Gainesville, Florida, USA. [4] Department of Surgical Oncology, University of Texas MD Anderson Cancer Center, Houston, Texas, USA. [5] Department of Genitourinary Medical Oncology, University of Texas MD Anderson Cancer Center, Houston, Texas, USA. *email: hnaora@mdanderson.org

t is widely recognized that the tumor microenvironment is modulated by cancer cells to support tumorigenesis, and that this stromal reprogramming is coordinated by direct cell-to-cell contact and/or soluble factors[1]. A classic example of this phenomenon is the orchestration by cancer-derived factors of endothelial cell growth, migration, and assembly into vessels that in turn supply oxygen and nutrients to sustain tumor growth[1]. Increasing evidence indicates that stromal reprogramming is also coordinated by extracellular vesicles (EVs) secreted by cancer cells. EVs are membranous structures that encapsulate biomolecular cargo such as RNA and proteins, and are often more highly secreted by cancer cells than by normal cells[2]. EVs vary in size and biogenesis. EVs include apoptotic bodies that are typically 1–5 μm in diameter and ectosomes that form through budding of the plasma membrane and range from 100 to 1000 nm in diameter[2,3]. Exosomes are EVs that derive from multivesicular endosomes and range from 30 to 150 nm in diameter[2,3]. Because of the difficulty in defining the sub-cellular origin of an EV, a size-based EV nomenclature has been recommended[3]. The most studied type of EV are small EVs (sEVs) that are < 200 nm in diameter[3]. sEVs that are secreted by cancer cells can suppress immune cell function[4,5] and can induce fibroblasts and mesenchymal stem cells to acquire an inflammatory phenotype[6,7]. Furthermore, several studies have shown that cancer cell-derived sEVs stimulate endothelial cell migration and vessel formation[7–10].

In a number of studies, the biological responses to sEVs have been attributed to constituents of their luminal cargo such as microRNAs and the activity of these constituents is contingent upon uptake of sEVs by recipient cells[6–10]. However, the significance of sEV-mediated RNA transfer has been questioned by studies that analyzed the stoichiometry of microRNAs and sEVs, and the fate of sEV RNA in recipient cells[11,12]. Furthermore, there is evidence that sEVs can deliver signals to various immune cells independently of uptake[4,13]. These findings suggest that sEVs might mediate intercellular communication in the tumor microenvironment through mechanisms other than transferring their luminal cargo into recipient cells. In this study, we identified that cancer cell-derived sEVs can stimulate endothelial cell migration and tube formation independently of uptake, and that these responses are mediated by the 189 amino acid, heparin-bound isoform of vascular endothelial growth factor (VEGF) that, unlike other common isoforms of VEGF, is enriched on the surface of sEVs. Furthermore, we found that sEV-associated VEGF (sEV-VEGF) is highly stable and is not neutralized by the therapeutic VEGF antibody bevacizumab, raising the possibility that elevated levels of sEV-VEGF contribute in part to the resistance of tumors to bevacizumab.

## Results

**Uptake-independent effects of sEVs on endothelial cells**. Unless noted otherwise, sEVs were isolated from culture media conditioned by ovarian, colorectal, and renal cancer cell lines, and from body fluids of tumor-bearing mice and cancer patients by sequential filtration to exclude particles of > 200 nm in diameter and soluble proteins of < 100 kDa in size, followed by density gradient ultracentrifugation. Details of sEV isolation are described in the Methods. TSG101, a marker that is enriched in sEVs[14], was exclusively detected in vesicles within the buoyant density range of sEVs (1.09–1.13 g/mL, termed "sEV fractions") (Fig. 1a and Supplementary Fig. 1). Vesicles in sEV fractions also expressed flotillin-1, CD63, and HSP70 (Fig. 1a, b and Supplementary Fig. 2a), which are expressed in sEVs but not exclusively[14]. We confirmed that vesicles in sEV fractions lacked α-actinin-4 and HSP90B1 (also known as GP96 or endoplasmin),

which are mainly expressed in larger EVs[14], and calnexin, a non-EV marker[3] (Supplementary Fig. 2a). Vesicles in sEV fractions were also evaluated for purity, size, and homogeneity by electron microscopy and nanoparticle tracking analysis, and were within the size range of sEVs (Fig. 1b and Supplementary Fig. 2b).

Following verification, sEVs were used to stimulate endothelial cells. sEVs induced cell migration and tube formation within 4–5 h (Fig. 1c–f). Uptake of sEVs was detected during this period but was modest (Supplementary Fig. 3a–d). To test whether sEVs can induce endothelial cell migration and tube formation independently of uptake, endothelial cells were treated with endocytosis inhibitors to block sEV uptake. As prolonged treatment with these inhibitors impairs cell motility, cells were treated for 4–5 h, and migrating cells and tubes counted thereafter. Blockade of sEV uptake was confirmed by fluorescence microscopy and flow cytometry (Fig. 1g, h and Supplementary Fig. 4a, b) and did not prevent sEVs from inducing endothelial cells to migrate and form tubes (Fig. 1c–f). These findings raise the possibility that cancer cell-derived sEVs can stimulate endothelial cell migration and tube formation via angiogenic proteins on the surface of sEVs rather than solely through transport of luminal cargo.

**VEGF is on the surface of cancer cell-derived sEVs**. In an unbiased effort to identify angiogenic factors associated with sEVs, we screened ovarian cancer cell-derived sEVs by using antibody (Ab) arrays. Several candidates including VEGF, growth-regulated oncogene-α (GROα), interleukin-8 (IL-8), and fibroblast growth factor-2 (FGF-2) were identified (Fig. 2a). Immunoassays were performed on lysates of sEVs secreted by ovarian (ES2), colorectal (HCT116), and renal (786-0) cancer cells, to determine the total amount of each factor in sEVs. Equivalent amounts of intact sEVs were assayed to determine the amount of each factor on the surface of sEVs. In control assays, no or minimal TSG101 (a luminal constituent of sEVs) was detected on intact sEVs, whereas the amount of CD63 (a surface protein) detected on intact sEVs was equivalent to the total CD63 content in sEVs (Fig. 2b). VEGF was detected on intact sEVs of all three cancer cell lines and at amounts equivalent to the total VEGF content in these sEVs (Fig. 2b). These findings indicate that almost all of the sEV-VEGF is present on the surface.

The presence of VEGF on the surface of sEVs was confirmed by two flow cytometric approaches. For negative controls, sEVs were isolated from a previously generated HCT116 VEGF[−/−] line[15] and from ES2 cells in which the VEGFA gene was deleted by CRISPR/Cas9 gene editing (Supplementary Fig. 5a–c). sEVs of isogenic VEGF[+/+] and VEGF[−/−] lines were similar in size and homogeneity (compare Supplementary Fig. 2b and 5d). In the first approach, microbeads were coupled to VEGF Ab, incubated with sEVs, and then stained with exo-FITC dye to label sEV membrane. Binding of Ab to VEGF on the surface of sEVs was evaluated by analyzing exo-FITC fluorescence in gated Ab-coupled microbeads. Gating strategy is shown in Supplementary Fig. 6a. Using this approach, VEGF was detected on the surface of VEGF[+/+] sEVs but not on VEGF[−/−] sEVs. Results were reproduced using three different VEGF Ab (Fig. 2c and Supplementary Fig. 6b, c). CD63 and TSG101 were assayed as positive and negative controls for sEV surface protein, respectively (Fig. 2c and Supplementary Fig. 6b, c). In the second approach, direct staining of sEVs with fluorochrome-conjugated Ab was evaluated in gated sEVs. Gating strategy is shown in Supplementary Fig. 7a, b. Using this approach, CD63 was detected on ~90% of both VEGF[+/+] and VEGF[−/−] sEVs, whereas VEGF was absent from VEGF[−/−] sEVs and detected on ~80% of VEGF[+/+] sEVs (Supplementary Fig. 7c, d). The presence

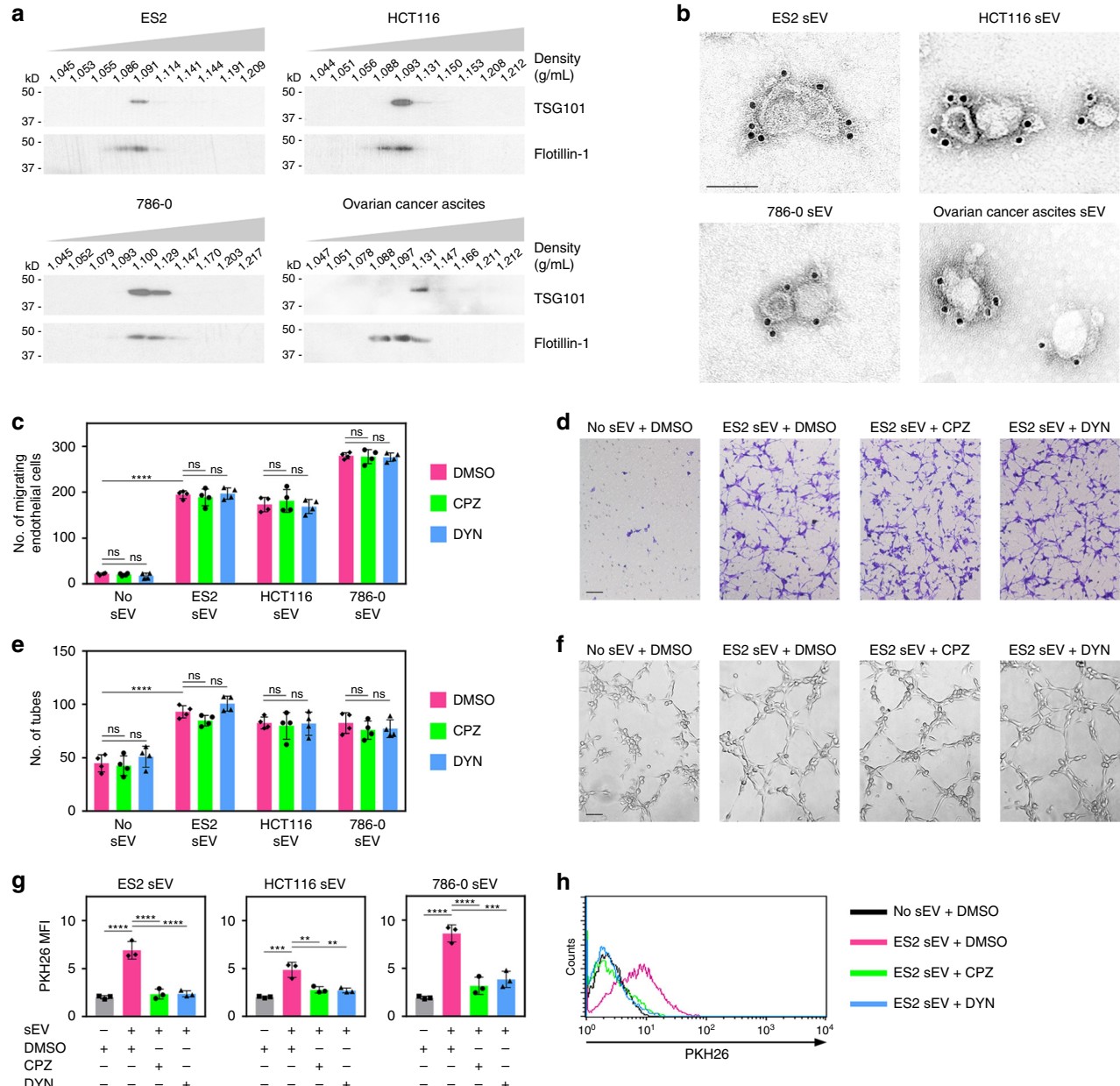

**Fig. 1 Cancer cell-derived sEVs can stimulate endothelial cell migration and tube formation independently of uptake. a** Immunoblot of TSG101 and flotillin-1 in fractions of the indicated buoyant densities that were isolated from media conditioned by ovarian (ES2), colorectal (HCT116), and renal (786-0) cancer cell lines, and from ovarian cancer patient ascites. **b** Immunogold labeling of CD63 on vesicles in sEV fractions (i.e., density of 1.09–1.13 g/mL). Scale bar = 100 nm. **c**–**f** Human umbilical vein endothelial cells (HUVEC) were pretreated with endocytosis inhibitors (chlorpromazine, CPZ; dynasore, DYN) or with dimethyl sulfoxide (DMSO) solvent, and then stimulated with sEVs of ES2, HCT116, and 786-0 cells. Shown are numbers (**c**) and representative images (**d**) of migrating HUVEC at 5 h after stimulation, and numbers (**e**) and representative images (**f**) of tubes formed at 4 h after stimulation. Mean ± SD of $n = 4$ independent experiments are shown. Scale bar = 100 μm. **g, h** HUVEC were treated as in **c** and evaluated for uptake of PKH26 dye-labeled sEVs by flow cytometry at 5 h thereafter. Shown are mean fluorescence intensity (MFI) values of PKH26 fluorescence detected in HUVEC in $n = 3$ independent experiments (**g**) and representative histogram plots (**h**). Gating strategy and contour plots are shown in Supplementary Fig. 3b and 4b, respectively. \*\**P* < 0.01, \*\*\**P* < 0.001, \*\*\*\**P* < 0.0001, by ANOVA with Bonferroni's corrections; one-way in **g**, two-way in **c** and **e**. ns: not significant. Source data used for graphs in **c**, **e**, and **g** can be found in Supplementary Data 1

of VEGF on the sEV surface was confirmed by immunogold labeling (Fig. 2d).

**sEV-VEGF is signaling competent**. VEGF binds to and activates three related tyrosine kinase receptors (VEGFRs), of which VEGFR2 mediates the majority of the angiogenic effects of VEGF[16,17]. Phosphorylation of VEGFR2 was induced in

endothelial cells following stimulation with cancer cell-derived sEVs (Fig. 3a and Supplementary Fig. 8). The sEV dose used (100 μg/mL) provided 500–2,000 pg/mL of sEV-VEGF (Fig. 2b). These concentrations of sEV-VEGF were within the range detected in body fluids of patients and mice with ovarian cancer (Table 1). As VEGF$_{165}$ is the most commonly overexpressed VEGF isoform in tumors[17], recombinant VEGF$_{165}$ was used as a positive control and at a concentration within the physiological

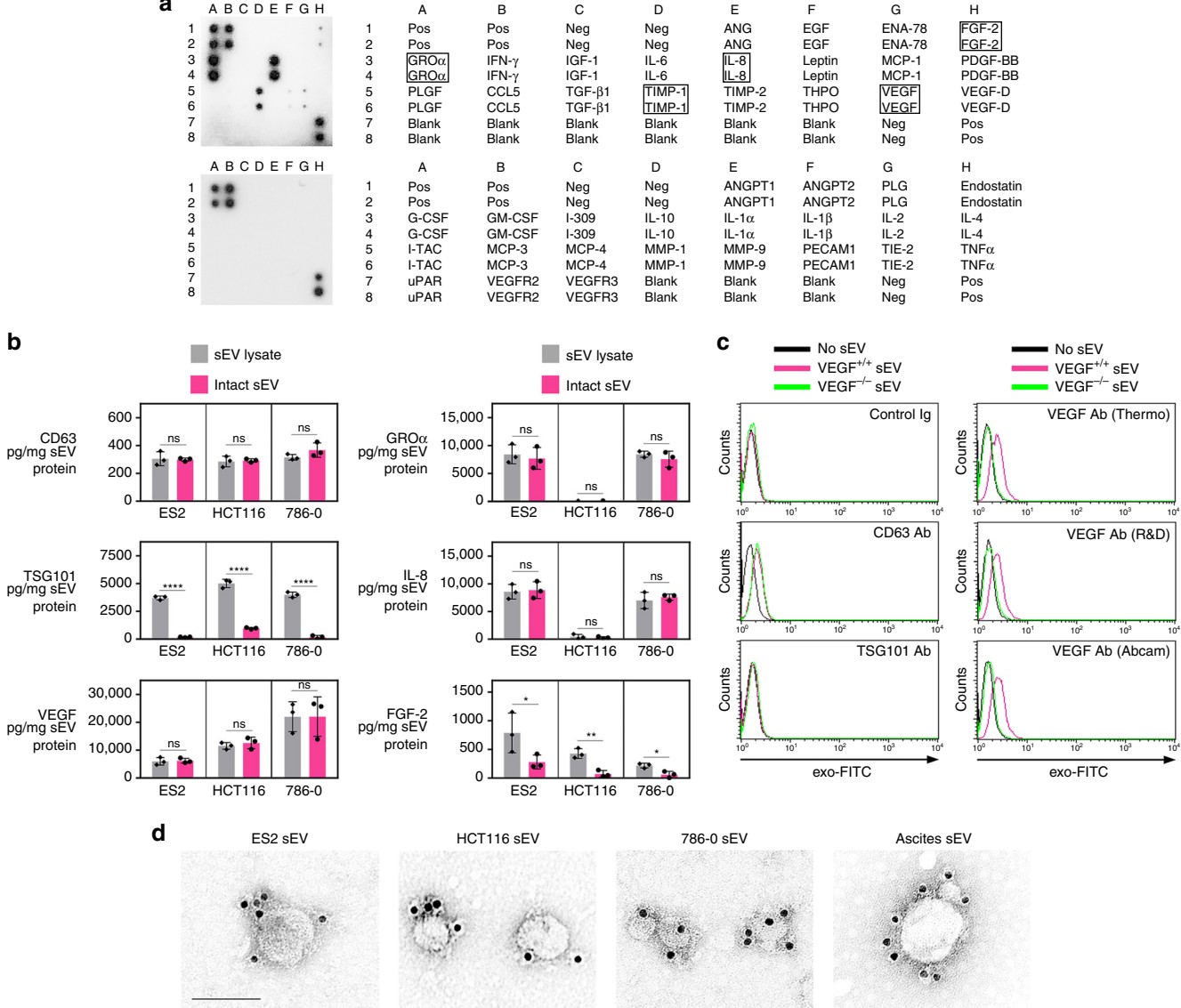

**Fig. 2** VEGF is present on the surface of cancer cell-derived sEVs. **a** Detection of angiogenesis-related proteins in sEVs of ES2 cells by Ab array. **b** Levels of angiogenic factors detected by enzyme-linked immunosorbent assays (ELISA) in lysates of sEVs (gray bars) and on the surface of equivalent amounts of intact sEVs (magenta bars) of ES2, HCT116, and 786-0 cells. CD63 and TSG101 were assayed as positive and negative controls for sEV surface protein, respectively. Shown are mean ± SD of $n = 3$ independent experiments. $*P < 0.05$, $**P < 0.01$, $****P < 0.0001$, by two-sided unpaired $t$-test. Source data can be found in Supplementary Data 2. **c** To detect sEV surface protein by flow cytometry, microbeads were coupled to the indicated Ab, incubated with sEVs of parental (VEGF$^{+/+}$) and VEGF-deficient (VEGF$^{-/-}$) HCT116 cells, and then stained with exo-FITC dye to label sEV membrane. Binding of Ab to protein on the surface of sEVs was evaluated by analyzing exo-FITC fluorescence in the gated population of Ab-coupled microbeads. Shown are representative histogram plots of fluorescence. Gating strategy, contour plots, and MFI values of $n = 3$ independent experiments are shown in Supplementary Fig. 6. **d** Immunogold labeling of VEGF on sEVs isolated from parental cancer cell lines and from ovarian cancer patient ascites. Scale bar = 100 nm

range (1000 pg/mL). The ability of sEVs to stimulate tube formation was abrogated when endothelial cells were treated with agents that inhibit VEGFR tyrosine kinase activity ($P < 0.0001$) (Fig. 3b, c) or with an Ab to VEGFR2 that blocks ligand binding ($P < 0.0001$) (Fig. 3d, e). These findings indicate that signaling-competent VEGF is present on the surface of cancer cell-derived sEVs, and that sEV-VEGF interacts with the extracellular domain of VEGFR2 on target cells.

We subsequently determined how much of the total VEGF in body fluids comprises sEV-VEGF. Samples of ascites from patients with ovarian cancer were depleted of sEVs or left non-depleted and then assayed for VEGF. The differences between

VEGF levels in whole and sEV-depleted samples revealed that 24–38% of the total VEGF in patient ascites comprises sEV-VEGF (Table 1). We also collected ascites from nude mice bearing human ovarian tumor xenografts (Supplementary Fig. 9a) and assayed whole and sEV-depleted samples for human (i.e., tumor-derived) VEGF. sEV-VEGF constituted 18–34% of tumor-derived VEGF in mouse ascites (Table 1). Analysis of purified sEVs by human- and mouse-specific VEGF immunoassays revealed that the vast majority of sEV-VEGF in mouse ascites derived from tumors and not the host (Supplementary Fig. 9b). Furthermore, analysis of VEGF levels in whole and sEV-depleted cancer cell-conditioned media revealed that sEV-VEGF

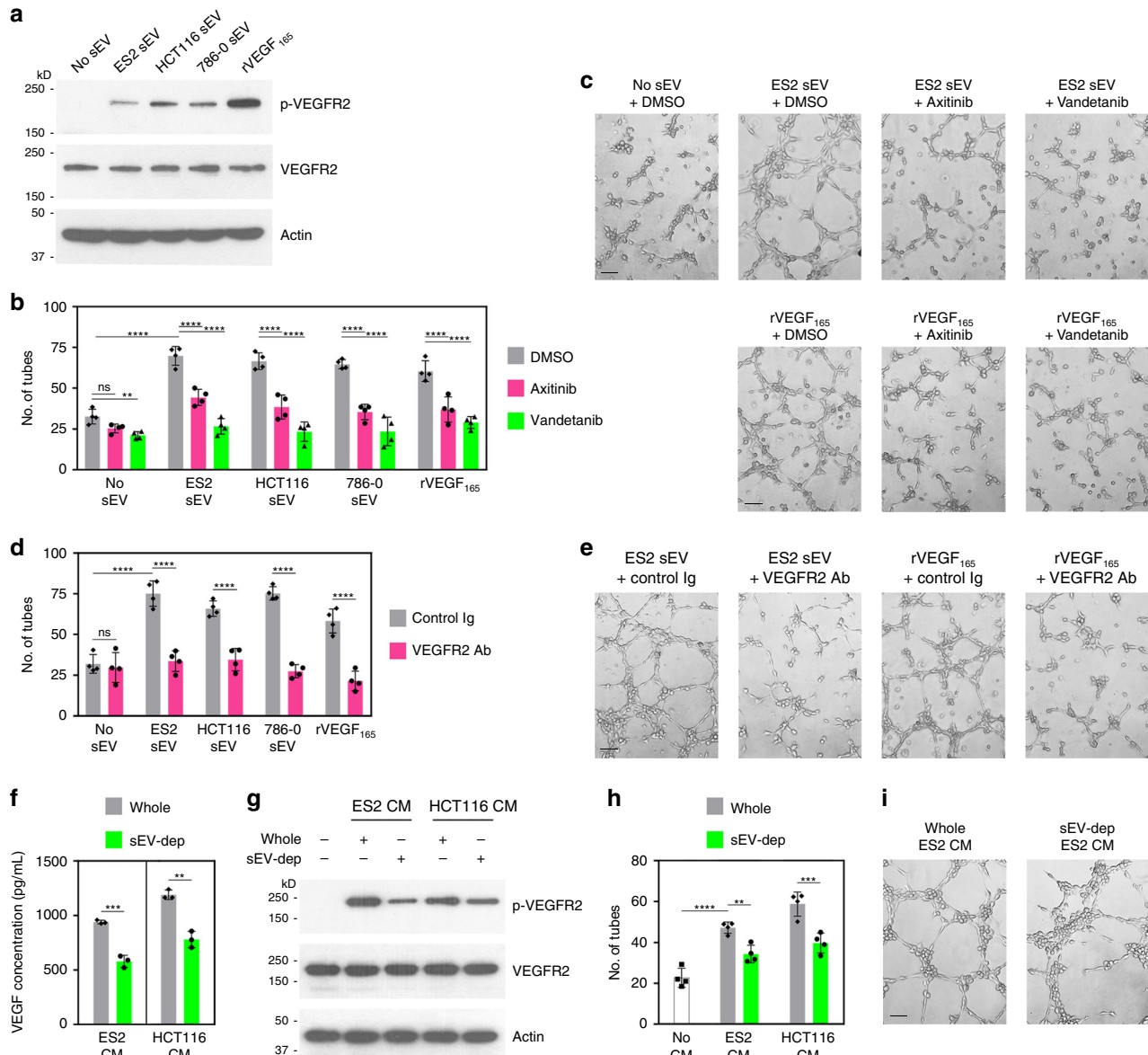

**Fig. 3** sEV-VEGF is signaling competent. **a** Immunoblot of phosphorylated VEGFR2 (p-VEGFR2) and total VEGFR2 in HUVEC at 5 min following stimulation with sEVs of ES2, HCT116, and 786-0 cells or with recombinant VEGF$_{165}$ (rVEGF$_{165}$). **b**–**e** HUVEC were pretreated with inhibitors of VEGFR tyrosine kinase activity (**b**, **c**) and with neutralizing Ab to VEGFR2 (**d**, **e**), stimulated with sEVs or rVEGF$_{165}$, and then assayed for tube formation at 4 h thereafter. In **b** and **d**, mean ± SD of $n = 4$ independent experiments. In **c** and **e**, representative images of tube formation. Scale bar = 100 μm. **f** VEGF levels in conditioned media (CM) of ES2 and HCT116 cells that were depleted of sEVs (sEV-dep) or left non-depleted (whole). Shown are mean ± SD of $n = 3$ independent experiments. **g**–**i** HUVEC were stimulated with whole and sEV-depleted conditioned media, and then assayed for VEGFR2 phosphorylation (**g**) and tube formation (**h**, **i**). In **h**, mean ± SD of $n = 4$ independent experiments. In **i**, representative images of tube formation. Scale bar = 100 μm. **P < 0.01, ***P < 0.001, ****P < 0.0001, by two-way ANOVA with Bonferroni's corrections in **b**, **d**, and **h**, and by a two-sided unpaired $t$-test in **f**. Source data used for graphs in **b**, **d**, **f**, and **h** can be found in Supplementary Data 3

constitutes ~35% of the total VEGF secreted by cancer cells (Fig. 3f). By comparing the effects of stimulating endothelial cells with whole and sEV-depleted media, we found that depletion of sEVs reduced VEGFR2 phosphorylation (Fig. 3g and Supplementary Fig. 8) and decreased tube formation by 30% (Fig. 3h, i).

**Effects of sEVs on endothelial cells depend on VEGF.** To determine whether the stimulatory effects of cancer cell-derived sEVs on endothelial cells are mediated by VEGF, we evaluated responses to sEVs of isogenic VEGF$^{+/+}$ and VEGF$^{-/-}$ cancer cells. In contrast to sEVs of VEGF$^{+/+}$ cells, sEVs of VEGF$^{-/-}$

cells neither induced VEGFR2 phosphorylation (Fig. 4a and Supplementary Fig. 8) nor significantly stimulated tube formation (Fig. 4b, c). To confirm our findings in vivo, nude mice were injected intraperitoneally (i.p.) with ES2 VEGF$^{-/-}$ cells and, at 1 week thereafter when tumors were palpable, were randomized into groups. Groups were administered equivalent amounts of sEVs of ES2 VEGF$^{+/+}$ cells or sEVs of ES2 VEGF$^{-/-}$ cells over the following 2 weeks. The sEV dose was determined from the volume of ascites and amount of tumor-derived sEV-VEGF in ascites that forms in mice at 3 weeks following i.p. injection of ES2 VEGF$^{+/+}$ cells (Supplementary Fig. 9a, b). When compared with saline-treated mice, tumor burden and numbers of

**Table 1 Abundance of sEV-VEGF in ascites of women and mice with ovarian cancer**

| | VEGF in whole ascites (pg/mL)[a] | VEGF in sEV-depleted ascites (pg/mL)[a] | Estimated sEV-VEGF in ascites (pg/mL)[b] |
|---|---|---|---|
| Ovarian cancer patient ascites[c] | | | |
| P1 | 2,313 | 1,435 | 878 (38.0%) |
| P2 | 1,098 | 768 | 330 (30.1%) |
| P3 | 4,056 | 3,098 | 958 (23.6%) |
| P4 | 1,837 | 1,362 | 475 (25.9%) |
| P5 | 1,930 | 1,379 | 551 (28.5%) |
| P6 | 1,118 | 708 | 410 (36.7%) |
| Ascites from mouse xenograft models[d] | | | |
| M1 | 12,909 | 10,532 | 2,377 (18.4%) |
| M2 | 12,302 | 9,230 | 3,072 (25.0%) |
| M3 | 13,807 | 11,025 | 2,782 (20.1%) |
| M4 | 8,724 | 5,727 | 2,997 (34.3%) |
| M5 | 13,187 | 9,925 | 3,262 (24.7%) |
| M6 | 10,962 | 8,876 | 2,086 (19.0%) |

[a]Assayed by ELISA. Shown is the mean of two independent assays of each ascites sample. Source data can be found in Supplementary Data 3
[b]Estimated from differences between VEGF levels in whole and sEV-depleted ascites samples. Proportion shown as % of VEGF in whole ascites in parentheses
[c]Clinical specimens of ascites from six women with Stage III high-grade serous ovarian carcinoma
[d]Ascites from six female nude mice, collected at 3 weeks following i.p. injection of ES2 human ovarian cancer cells

intratumoral endothelial cells were increased in mice that had been administered $VEGF^{+/+}$ sEVs ($P < 0.0001$) but not in mice that had been administered $VEGF^{-/-}$ sEVs (Fig. 4d–f). VEGF induces ascites by stimulating vascular permeability[18]. Notably, ascites was induced in mice by $VEGF^{+/+}$ sEVs ($P < 0.001$) but not by $VEGF^{-/-}$ sEVs (Fig. 4f). These findings demonstrate that sEV-VEGF is biologically active in vivo, and that the stimulatory effects of cancer cell-derived sEVs on endothelial cells and tumor growth depend on VEGF.

**sEV-VEGF predominantly comprises dimeric $VEGF_{189}$.** VEGF has been detected in cancer cell-derived sEVs[8,19,20], but the molecular characteristics of the sEV form of VEGF and the mechanism by which VEGF associates with sEVs have not been delineated. Alternative splicing of *VEGFA* mRNA yields several VEGF isoforms of which the 121, 165, 189, and 206 amino acid variants are the most common[16]. $VEGF_{121}$ and the other common isoforms all contain exons 1 to 5 and exon 8, and the larger isoforms additionally contain exons 6 and/or 7 that encode heparin-binding domains[16]. $VEGF_{121}$ is freely secreted, $VEGF_{189}$ and $VEGF_{206}$ are membrane-bound, and $VEGF_{165}$ exists in both soluble and membranous forms[16]. All of the VEGF isoforms are biologically active as homodimers[21]. Monomers of $VEGF_{121}$ and $VEGF_{165}$, and dimers of $VEGF_{121}$, $VEGF_{165}$, and $VEGF_{189}$ were detected at various ratios in cells of ovarian, colorectal, and renal cancer lines (Fig. 5a and Supplementary Fig. 10). In contrast, sEVs secreted by these cells were enriched with $VEGF_{189}$ dimers (Fig. 5b and Supplementary Fig. 10). To eliminate the possibility that the presence of VEGF resulted from contamination during ultracentrifugation, we assayed all fractions for VEGF. VEGF was detected in the highest density fractions that largely consisted of unfractionated and/or soluble material, and this VEGF comprised $VEGF_{121}$ and $VEGF_{165}$ but not $VEGF_{189}$ (Supplementary Fig. 11a, b). Of the other fractions, only the fractions of the density of sEVs showed prominent levels of VEGF and this VEGF comprised dimeric $VEGF_{189}$ (Supplementary Fig. 11a, b). To confirm that $VEGF_{189}$ is preferentially enriched in sEVs, we evaluated clinical specimens. Multiple isoforms of VEGF were detected at various ratios in ovarian tumor tissues, but dimeric $VEGF_{189}$ was the predominant species in sEVs isolated from body fluids of the same patients (Fig. 5c and Supplementary Fig. 10). $VEGF_{189}$ was

also the most abundant isoform of VEGF in sEVs isolated from body fluids of patients with colorectal or renal cancers (Fig. 5d and Supplementary Fig. 10).

**sEV-VEGF is heparin-bound and highly stable.** To confirm that $VEGF_{189}$ preferentially localizes to sEVs, we reconstituted $VEGF_{189}$ and two other major isoforms of VEGF ($VEGF_{121}$ and $VEGF_{165}$) individually into $VEGF^{-/-}$ cancer cells, and then assayed the VEGF content in sEVs secreted by these cells. sEVs of cells that expressed $VEGF_{189}$ had the highest VEGF content (Fig. 6a). Analysis of VEGF levels in whole and sEV-depleted conditioned media of $VEGF_{189}$-transfected $VEGF^{-/-}$ cells revealed that nearly 90% of the $VEGF_{189}$ was sEV-associated (Fig. 6b). By contrast, analysis of whole and sEV-depleted conditioned media of $VEGF_{121}$-transfected $VEGF^{-/-}$ cells indicated that almost all of the $VEGF_{121}$ was not sEV-associated (Fig. 6c). To test whether $VEGF_{189}$ binds to sEVs following secretion as opposed to being sorted into sEVs, we incubated conditioned media of non-transfected $VEGF^{-/-}$ cells (which contains secreted EVs but no VEGF) with recombinant $VEGF_{189}$ (i.e., "free $VEGF_{189}$"), which was added at an amount equivalent to the total amount of VEGF secreted by $VEGF_{189}$-transfected $VEGF^{-/-}$ cells. Thereafter, sEVs were isolated from the media. The amount of $VEGF_{189}$ detected in these sEVs was ~25% of the amount of $VEGF_{189}$ in sEVs secreted by $VEGF_{189}$-transfected $VEGF^{-/-}$ cells (Fig. 6d). These findings suggest that, although $VEGF_{189}$ can bind to sEVs post secretion, the presence of this ligand in sEVs predominantly occurs through selective sorting into sEVs.

$VEGF_{121}$ lacks a heparin-binding domain, whereas $VEGF_{189}$ has substantially higher affinity for heparin than $VEGF_{165}$[16]. As sEVs contain membrane-associated heparan sulfate proteoglycans[22], we investigated the possibility that heparan sulfate mediates localization of $VEGF_{189}$ in sEVs. We overexpressed human $VEGF_{189}$ in CHO-K1 cells and in pgsD-677 cells, a CHO cell mutant that is deficient in heparan sulfate biosynthesis[23], and then evaluated sEVs secreted by these cells. Whereas total levels of exogenous $VEGF_{189}$ did not differ between CHO-K1 and pgsD-677 cells, sEVs of pgsD-677 cells contained substantially less $VEGF_{189}$ than sEVs of CHO-K1 cells ($P < 0.01$) (Fig. 6e). To test whether heparan sulfate facilitates the interaction of $VEGF_{189}$ with the surface of sEVs, we evaluated the presence of VEGF following incubation of cancer cell-derived sEVs with heparinase, an enzyme that cleaves heparan sulfate chains. VEGF was removed from sEVs by treatment with heparinase but not with chondroitinase that degrades chondroitin sulfate chains (Fig. 6f, g and Supplementary Fig. 12). These findings indicate that $VEGF_{189}$ interacts with the surface of sEVs, at least in part, via heparin-binding. Although VEGF has been detected in sEVs[8,19,20], the possibility that association with sEVs changes properties of the ligand has not been investigated. Following incubation in human plasma at 37 °C, the levels of recombinant $VEGF_{189}$ rapidly declined and were undetectable at 24 h (Fig. 6h). By contrast, sEV-VEGF was substantially more stable and its levels remained almost unchanged at 24 h following incubation in the plasma (Fig. 6h). These findings indicate that association of $VEGF_{189}$ with the surface of sEVs profoundly increases the half-life of the ligand.

**sEV-VEGF is not neutralized by bevacizumab in vitro.** Bevacizumab is a humanized monoclonal Ab that recognizes all isoforms of human VEGF and is the most studied anti-angiogenic agent[24]. Whereas binding of bevacizumab to soluble isoforms of VEGF has been characterized[25], the ability of bevacizumab to neutralize VEGF when associated with other proteins is poorly

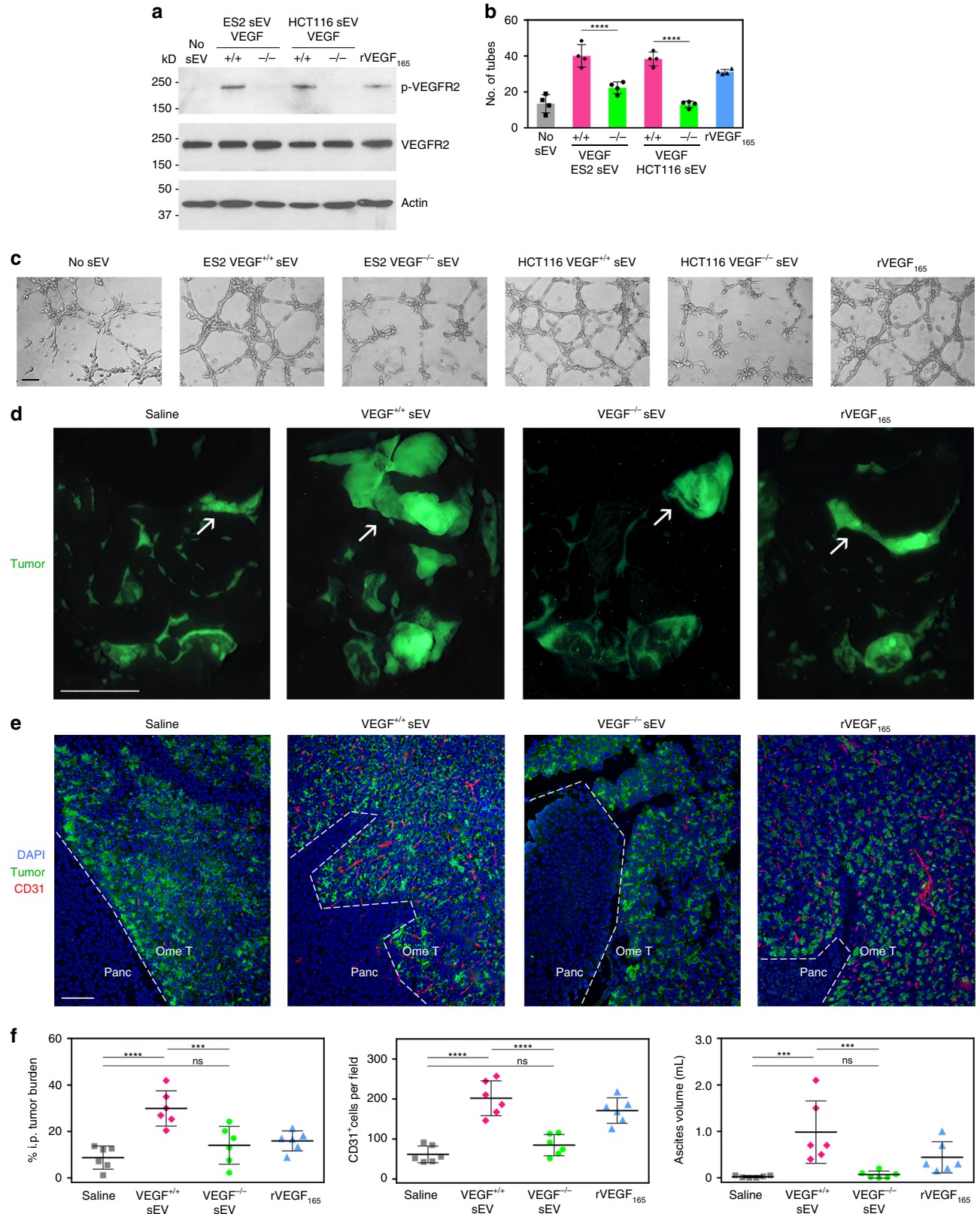

understood. Binding of bevacizumab to $VEGF_{189}$ was significantly reduced when $VEGF_{189}$ was pre-bound to high-molecular-weight (HMW) heparin ($P < 0.0001$) (Fig. 7a, b). As $VEGF_{189}$ associates with sEVs via heparin-binding (Fig. 6f, g), we tested whether bevacizumab binds sEV-VEGF. We initially confirmed that

recognition of VEGF by bevacizumab can be detected by flow cytometry, and that bevacizumab can be coupled to microbeads (Supplementary Fig. 13a–d). Bevacizumab-coupled microbeads were then incubated with sEVs, followed by labeling of sEV membrane with exo-FITC dye. Binding of bevacizumab to

**Fig. 4 Stimulatory effects of cancer cell-derived sEVs on endothelial cells and tumor growth depend on VEGF. a–c** HUVEC were stimulated with equivalent amounts of sEVs of VEGF$^{+/+}$ cancer cells, sEVs of VEGF$^{-/-}$ cancer cells or rVEGF$_{165}$, and then assayed for VEGFR2 phosphorylation (**a**) and tube formation (**b, c**). In **b**, mean ± SD of $n = 4$ independent experiments. In **c**, representative images of tube formation. Scale bar = 100 μm. **d–f** Nude mice were inoculated i.p. with ES2 VEGF$^{-/-}$ cells that stably expressed GFP. At 7 days thereafter when tumors were palpable, mice were randomized into groups ($n = 6$ mice per group) and then administered equivalent amounts of sEVs of ES2 VEGF$^{+/+}$ cells or sEVs of ES2 VEGF$^{-/-}$ cells, three times a week for 2 weeks. Negative and positive control groups of tumor-bearing mice were administered saline and rVEGF$_{165}$, respectively. In **d**, representative images of GFP-expressing tumors in the abdominal cavity viewed under a fluorescence stereomicroscope. Arrows indicate tumors on the omentum. Scale bar = 10 mm. In **e**, immunofluorescence staining of CD31 (red) in sections of omental tumors (Ome T) adjacent to the pancreas (Panc). Scale bar = 100 μm. In **f**, amount of i.p. tumor burden, numbers of intratumoral CD31$^+$ cells, and volume of ascites in each mouse in each of the groups. I.p. tumor burden is expressed as % of area of GFP fluorescence in the abdominal cavity. Numbers of CD31$^+$ cells were scored in five random 100× fields per omental tumor section and an average score was determined for each mouse. Error bars in **f** represent SD. ***$P < 0.001$, ****$P < 0.0001$, by one-way ANOVA with Bonferroni's corrections in **b** and **f**. Source data used for graphs in **b** and **f** can be found in Supplementary Data 4

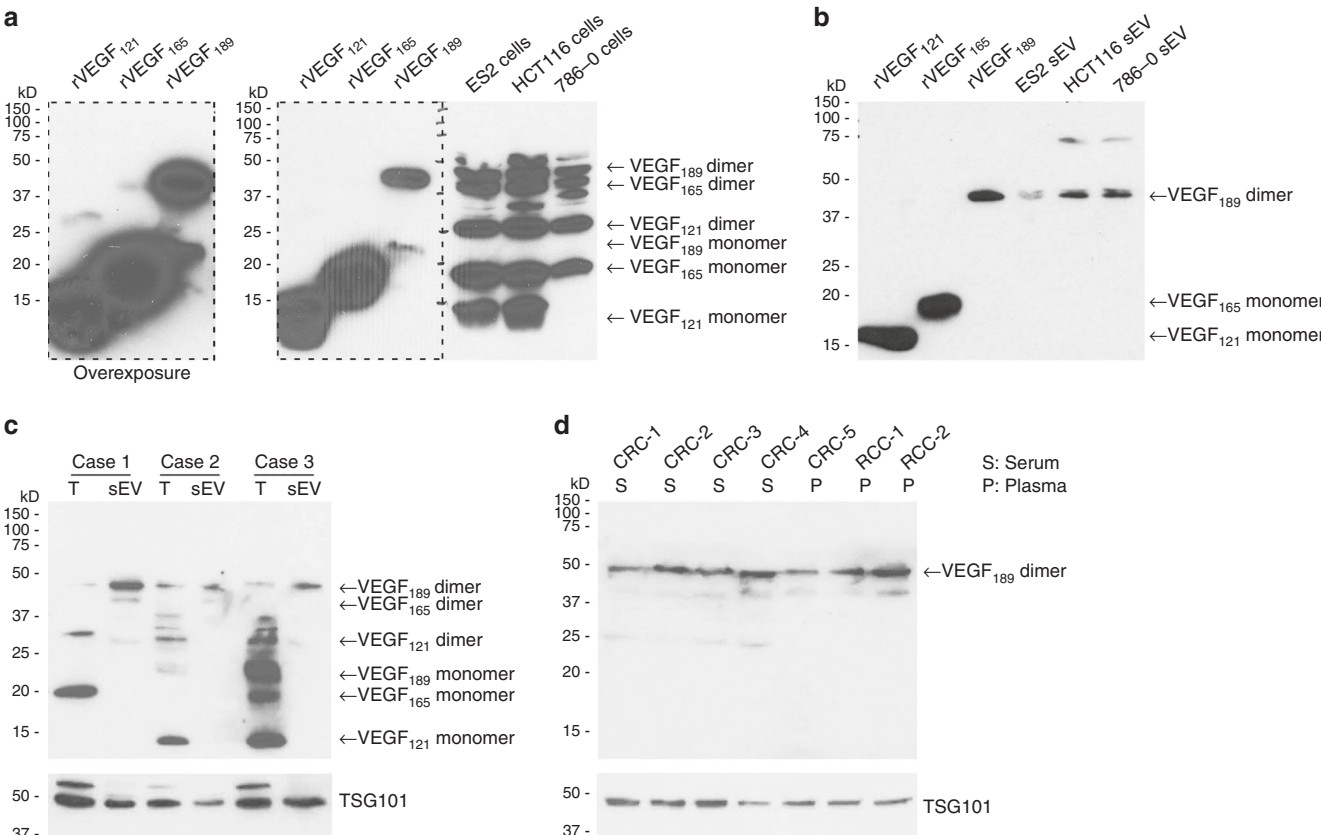

**Fig. 5 sEV-VEGF predominantly comprises dimeric VEGF$_{189}$. a** Immunoblot of cellular VEGF in lysates of cells of parental cancer cell lines that were treated with brefeldin A to block protein secretion. Recombinant VEGF proteins were included as controls. Overexposure shows VEGF$_{121}$ and VEGF$_{165}$ dimers. **b** Immunoblot of VEGF in sEVs isolated from the same cell lines as in **a** but without brefeldin A treatment. **c** Immunoblot of VEGF in tumor tissues (T) of three patients with ovarian cancer and in sEVs isolated from ascites of the same patients. **d** Immunoblot of VEGF in sEVs isolated from serum or plasma of five patients with colorectal carcinoma (CRC) and two patients with renal cell carcinoma (RCC). TSG101 was assayed as a control in **c, d**

sEV-VEGF was evaluated by analyzing exo-FITC fluorescence on coupled microbeads. In control assays, sEVs were incubated with microbeads coupled to VEGFR1/R2-Fc, a chimera that consists of the ligand-binding domains of VEGFR1 and VEGFR2 fused to the Fc portion of human IgG$_1$. VEGFR1/R2-Fc bound to VEGF$^{+/+}$ sEVs but not VEGF$^{-/-}$ sEVs (Fig. 7c, d and Supplementary Fig. 14a). This finding was consistent with the activation of VEGFR2 by VEGF$^{+/+}$ sEVs and not VEGF$^{-/-}$ sEVs (Fig. 4a). In contrast, bevacizumab did not bind to VEGF$^{+/+}$ sEVs (Fig. 7c, d and Supplementary Fig. 14a).

To confirm our findings, we incubated bevacizumab with either recombinant VEGF or VEGF$^{+/+}$ sEVs and thereafter assayed the levels of unbound bevacizumab. Levels of unbound bevacizumab decreased following incubation with increasing amounts of recombinant VEGF but did not decrease following incubation with VEGF$^{+/+}$ sEVs containing equivalent amounts of VEGF (Fig. 7e). Consistent with these findings, bevacizumab blocked VEGFR2 phosphorylation and tube formation in endothelial cells that were stimulated with recombinant VEGF but not in cells stimulated with VEGF$^{+/+}$ sEVs (Fig. 7f–h and Supplementary Fig. 8). To test the neutralizing ability of bevacizumab under conditions where soluble VEGF and sEVs carrying VEGF are co-secreted, cancer cell-conditioned media was depleted of sEVs or left whole, and then incubated with bevacizumab. Following incubation, levels of unbound bevacizumab in whole media were almost identical to those in sEV-depleted media (Supplementary Fig. 14b), indicating that bevacizumab only neutralized non-sEV-VEGF. This was

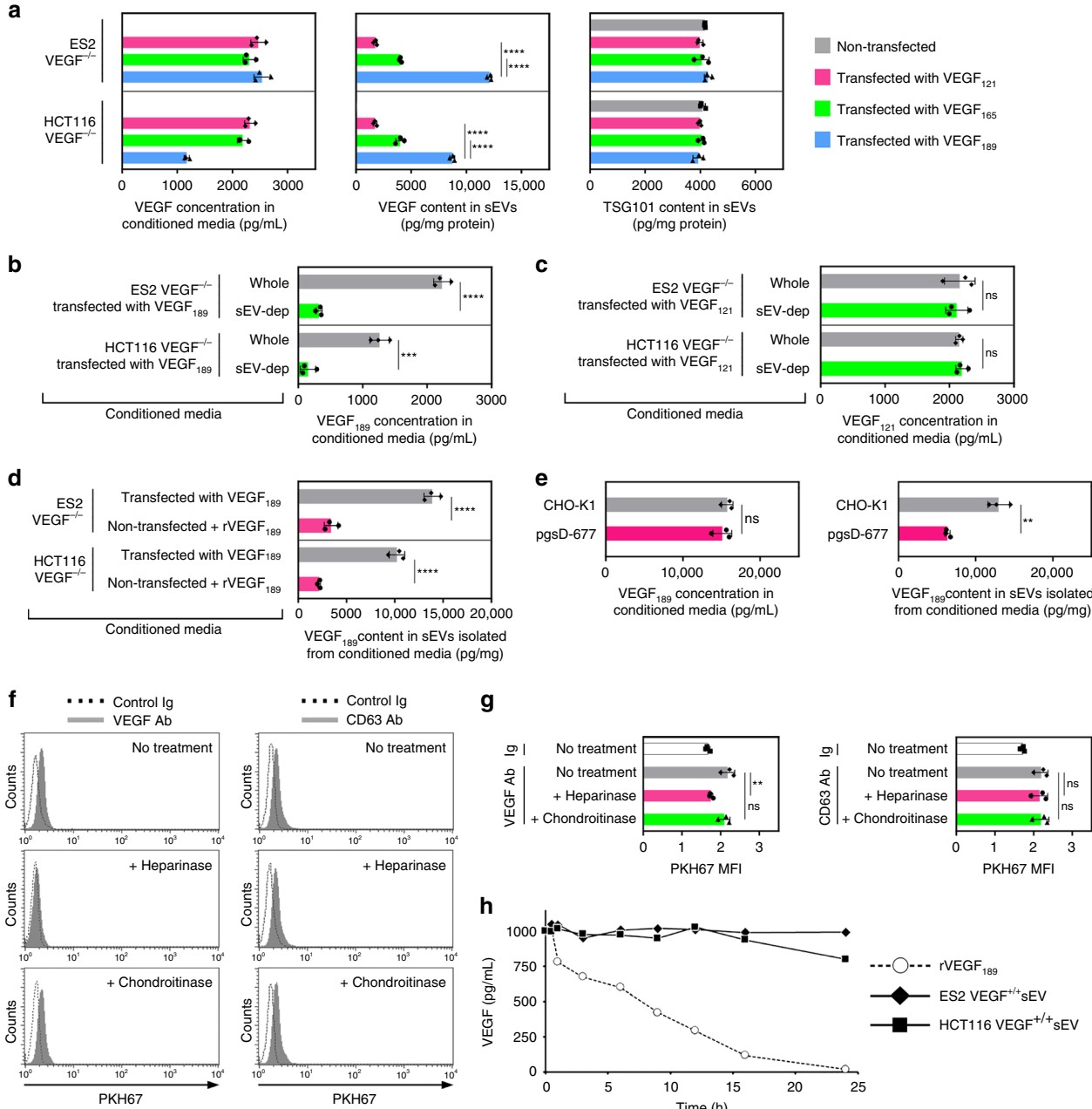

**Fig. 6 Selective localization of VEGF$_{189}$ in sEVs is mediated by heparin-binding and increases ligand stability. a** VEGF levels in conditioned media and sEVs of ES2 VEGF$^{-/-}$ and HCT116 VEGF$^{-/-}$ cells transfected with VEGF$_{121}$, VEGF$_{165}$, or VEGF$_{189}$. TSG101 was assayed in sEVs as a control. Mean ± SD of $n = 3$ independent experiments are shown. **b, c** VEGF levels in whole and sEV-depleted conditioned media of VEGF$^{-/-}$ cells transfected with VEGF$_{189}$ (**b**) and VEGF$_{121}$ (**c**). Mean ± SD of $n = 3$ independent experiments are shown. **d** Conditioned media of non-transfected VEGF$^{-/-}$ cells was incubated with addition of recombinant VEGF$_{189}$ (rVEGF$_{189}$) at a concentration equivalent to the VEGF concentration in conditioned media of VEGF$_{189}$-transfected VEGF$^{-/-}$ cells (2500 pg/mL for ES2; 1500 pg/mL for HCT116; see data in **a**). Thereafter, sEVs were isolated. Amounts of VEGF$_{189}$ detected in these sEVs were compared with VEGF content in sEVs secreted by VEGF$_{189}$-transfected VEGF$^{-/-}$ cells. Mean ± SD of $n = 3$ independent experiments are shown. **e** Levels of human VEGF$_{189}$ in conditioned media and sEVs of CHO-K1 and pgsD-677 cells transfected with human VEGF$_{189}$. Mean ± SD of $n = 3$ independent experiments are shown. **f, g** PKH67-labeled sEVs of parental ES2 cells were pretreated with heparinase, chondroitinase, or no enzyme, and then incubated with VEGF Ab coupled to microbeads. VEGF on sEVs was detected by flow cytometric analysis of PKH67 fluorescence in Ab-coupled microbeads (solid histograms). Dotted histograms show background fluorescence when sEVs were incubated with control Ig-coupled beads. As a negative control for enzymatic digestion, the same approach was used to detect the transmembrane protein CD63. In **f**, representative histogram plots. In **g**, MFI values of $n = 3$ independent experiments (mean ± SD). Contour plots are shown in Supplementary Fig. 12. **h** rVEGF$_{189}$ and sEVs with an equivalent content of VEGF were added to healthy donor plasma. Following incubation at 37 °C for the indicated times, VEGF levels in plasma were assayed. Shown are mean of $n = 2$ independent experiments. **$P < 0.01$, ***$P < 0.001$, ****$P < 0.0001$, by one-way ANOVA with Bonferroni's corrections in **a** and **g**, by two-sided unpaired $t$-test in **b–e**. Source data used for graphs in **a–e**, **g**, and **h** can be found in Supplementary Data 5

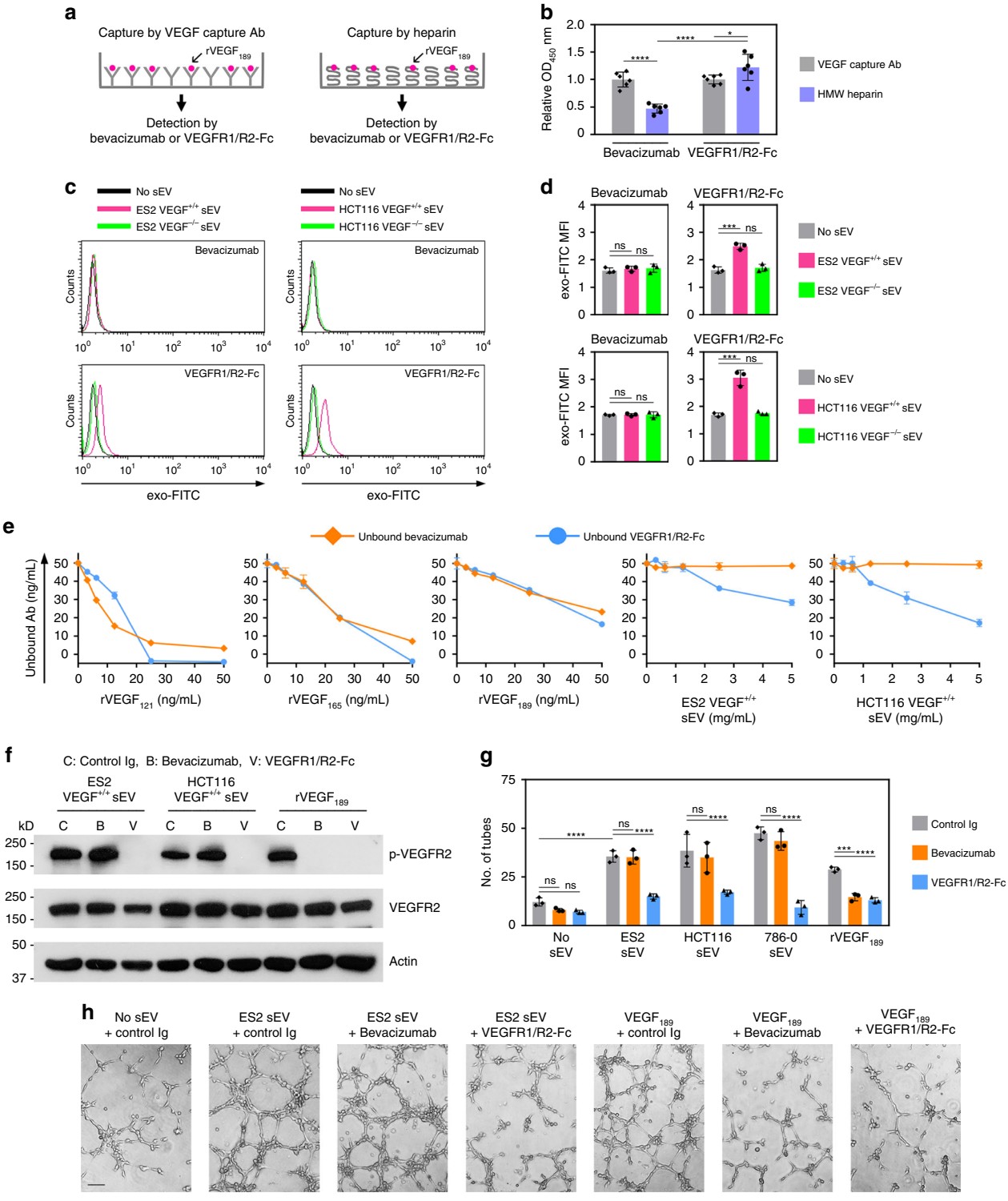

confirmed by the decrease in unbound bevacizumab levels following incubation in sEV-depleted media to which recombinant $VEGF_{189}$ had been added (Supplementary Fig. 14b). Consistent with the ability of bevacizumab to bind non-sEV-VEGF and not sEV-VEGF, bevacizumab completely blocked VEGFR2 phosphorylation in endothelial cells that were stimulated with sEV-depleted media but only partially inhibited VEGFR2 phosphorylation in endothelial cells stimulated with whole media (Supplementary Fig. 14c).

**sEV-VEGF is not neutralized by bevacizumab in vivo.** To confirm our findings in vivo, we injected nude mice i.p. with ES2 $VEGF^{-/-}$ cells and thereafter randomized mice into groups that were then administered sEVs of ES2 $VEGF^{+/+}$ cells with bevacizumab or control Ig, or administered recombinant $VEGF_{189}$ with bevacizumab or control Ig. As compared with treatment with control Ig, bevacizumab inhibited tumor growth ($P < 0.001$), angiogenesis ($P < 0.0001$), and ascites ($P < 0.001$) in mice that had been administered recombinant $VEGF_{189}$, but not in mice that

**Fig. 7** Heparin-bound sEV-VEGF is not neutralized by bevacizumab in vitro. **a, b** rVEGF$_{189}$ was captured by HMW heparin or by VEGF capture Ab (positive control), and then incubated with bevacizumab or VEGFR1/R2-Fc. Bevacizumab bound to VEGF$_{189}$ and VEGFR1/R2-Fc bound to VEGF$_{189}$ were detected by anti-human IgG. In **a**, experimental scheme. In **b**, relative levels of bevacizumab bound to VEGF$_{189}$ and VEGFR1/R2 bound to VEGF$_{189}$. Shown are mean ± SD of $n = 6$ independent experiments. **c, d** Microbeads were coupled to bevacizumab, incubated with sEVs of VEGF$^{+/+}$ cells or sEVs of VEGF$^{-/-}$ cells, and then stained with exo-FITC dye to label sEV membrane. The same procedure was performed using microbeads coupled to VEGFR1/R2-Fc (positive control). Binding of bevacizumab and VEGFR1/R2-Fc to VEGF on the surface of sEVs was evaluated by flow cytometric analysis of exo-FITC fluorescence in the gated population of microbeads. In **c**, representative histogram plots. In **d**, MFI values of $n = 3$ independent experiments (mean ± SD). Gating strategy is shown in Supplementary Fig. 6a. Contour plots are shown in Supplementary Fig. 14a. **e** Bevacizumab and VEGFR1/R2-Fc were incubated with recombinant VEGF and with sEVs that have a VEGF content equivalent to the range of amounts of recombinant VEGF. Following incubation, levels of unbound bevacizumab and unbound VEGFR1/R2-Fc were assayed. Shown are mean ± SD of $n = 3$ independent experiments. **f–h** HUVEC were stimulated with sEVs or rVEGF$_{189}$ that were pre-incubated with control Ig, bevacizumab, or VEGFR1/R2-Fc, and then assayed for phosphorylated and total VEGFR2 (**f**) and tube formation (**g, h**). In **g**, mean ± SD of $n = 3$ independent experiments. In **h**, representative images of tube formation. Scale bar = 100 μm. $*P < 0.05$, $***P < 0.001$, $****P < 0.0001$ by ANOVA with Bonferroni's corrections; one-way in **d**; two-way in **b** and **g**. Source data used for graphs in **b, d, e,** and **g** can be found in Supplementary Data 6

had been administered VEGF$^{+/+}$ sEVs (Fig. 8a–e). We next evaluated the effect of bevacizumab on endogenous levels of tumor-derived sEV-VEGF and non-sEV-VEGF by treating mice bearing subcutaneous (s.c.) tumors derived from parental ES2 and HCT116 cells with bevacizumab for 1 week. Bevacizumab did not inhibit tumor growth during this period (Supplementary Fig. 15a), as observed in other studies that used the same models[26,27]. Plasma samples collected from mice pre- and post-treatment, were depleted of sEVs or left non-depleted and then assayed for human VEGF. sEV-VEGF levels were estimated from differences between VEGF levels in whole and sEV-depleted plasma (Supplementary Fig. 15b–d). Following bevacizumab treatment, non-sEV-VEGF levels decreased (ES2, $P < 0.05$; HCT116, $P < 0.001$), whereas sEV-VEGF levels increased (ES2, $P < 0.01$; HCT116, $P < 0.05$) (Supplementary Fig. 15c, d).

**Clinical significance of high baseline levels of sEV-VEGF.** Our findings that bevacizumab does not neutralize sEV-VEGF raise the possibility that cancer patients who have elevated levels of sEV-VEGF might not benefit from bevacizumab. Bevacizumab has been approved for a variety of solid tumors in combination with chemotherapy and/or in a recurrent setting[28]. As combination therapy and prior treatment can complicate interpretations, we investigated the relationship between baseline levels of sEV-VEGF and outcomes in a cohort of newly diagnosed cancer patients who received bevacizumab monotherapy. Specifically, we analyzed plasma samples from a Phase II trial of patients with newly diagnosed Stage IV metastatic renal cell carcinoma, who were treated presurgically with single-agent bevacizumab and thereafter restaged (study NCT00113217)[29]. As volumes of these plasma samples were too small for sEVs to be isolated by density gradient ultracentrifugation, sEVs were isolated by using Exo-Quick reagent. We confirmed that these sEVs expressed sEV markers, lacked markers of larger EVs and non-EV components, and had a size distribution similar to that of sEVs isolated by density gradient ultracentrifugation (Supplementary Figs. 16a, b and 17).

Baseline levels of total VEGF and sEV-VEGF were assayed in plasma samples blinded to clinical data and thereafter evaluated for relationships with outcomes. No significant difference in baseline total VEGF levels was found between patients who showed disease progression following bevacizumab treatment and patients who had stable or regressing disease (Fig. 8f). In contrast, baseline sEV-VEGF levels were ~5-fold higher in patients with progressing disease than in patients with stable or regressing disease ($P = 0.010$) (Fig. 8g). These findings suggest that baseline levels of sEV-VEGF are more informative for bevacizumab treatment benefit than levels of total VEGF.

## Discussion

sEV-mediated intercellular communication has been described in diverse contexts including the tumor microenvironment and was predominantly thought to occur through the uptake of vesicular cargo by recipient cells[2,6–10]. However, recent studies have shown that sEVs can also signal to recipient cells via proteins on the vesicular surface such as PD-L1 and E-cadherin[5,30]. Several growth factors including VEGF have been detected in cancer cell-derived sEVs and were assumed to be luminal constituents[7,8]. VEGF has been implicated in the angiogenic activity of these sEVs[19,20], but there has been no explanation as to how this ligand elicits its signals if it is encapsulated within sEVs and then internalized in recipient cells. Here we identified that cancer cell-derived sEVs can stimulate endothelial cell migration and tube formation independently of uptake, and that these responses are mediated by heparin-bound VEGF on the surface of sEVs. Our findings that sEV-VEGF is signaling-competent is consistent with a report that heparin-bound VEGF can activate VEGFR2 phosphorylation, and that this activation does not require VEGF internalization[31]. It is increasingly recognized that the molecular composition of sEVs is distinct from the cellular profile[2] but little is known as to whether and how highly related proteins are differentially localized to sEVs. Our study shows that sEV-VEGF predominantly comprises VEGF$_{189}$ in its active dimeric form, and that VEGF$_{189}$, but not two other major isoforms (VEGF$_{121}$ and VEGF$_{165}$), preferentially localizes to sEVs. VEGF$_{189}$ has substantially higher affinity for heparin than VEGF$_{165}$ and VEGF$_{121}$ does not bind heparin[16]. The notion that VEGF$_{189}$ preferentially localizes to sEVs, at least in part, because of its high affinity for heparin is supported by our findings that the VEGF$_{189}$ content is substantially reduced in sEVs of cells that are deficient in heparan sulfate biosynthesis, and that heparinase removes VEGF$_{189}$ from the surface of sEVs (Fig. 6e–g). Although we found that VEGF$_{189}$ can bind to sEVs post secretion, post-secretion binding only partially explained the presence of VEGF$_{189}$ in sEVs (Fig. 6d). Intriguingly, Feng et al.[32] identified that breast cancer cell-derived large EVs (lEV) of 0.5–1.0 μm in diameter contain a 90 kDa form of VEGF (VEGF$_{90K}$) that comprises crosslinked VEGF$_{165}$ bound to heat shock protein 90[32]. Notably, the authors neither detected other forms of VEGF in lEVs nor detected VEGF$_{90K}$ in breast cancer sEVs[32]. Similarly, we did not detect VEGF$_{90K}$ in sEVs isolated from other cancer cell types and from body fluids of cancer patients (Fig. 5b–d). The differences between our findings in sEVs and those of Feng et al.[32] in lEVs implicate that different isoforms of VEGF are sorted by distinct mechanisms, resulting in compartmentalization into different types of EVs.

Our finding that VEGF associates with the sEV surface via heparin binding raises the possibility that other growth factors that bind heparin, such as GROα, IL-8, and FGF-2[33–35], are

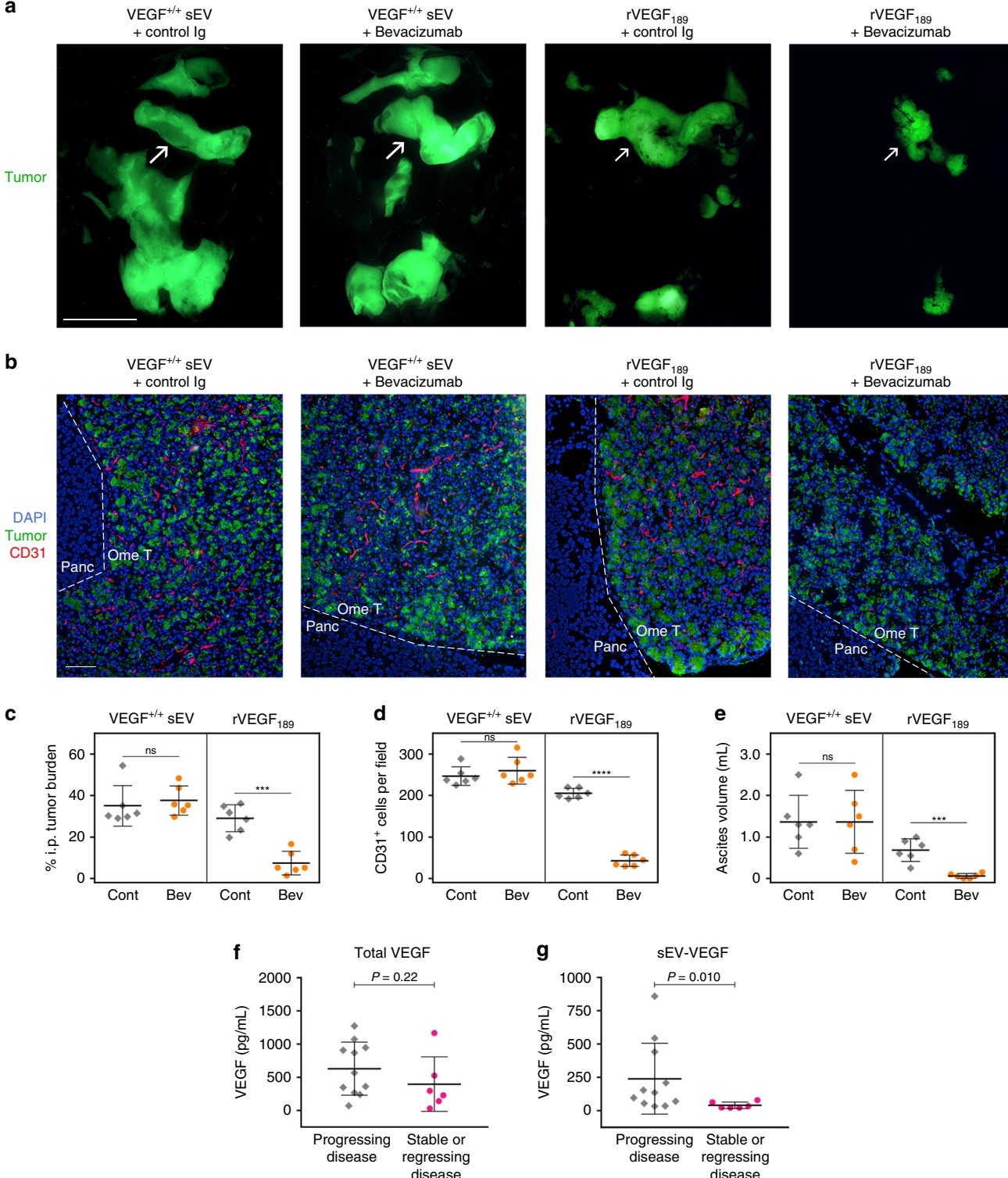

**Fig. 8** sEV-VEGF is not neutralized by bevacizumab in vivo and is associated with disease progression in bevacizumab-treated cancer patients. **a–e** Nude mice were inoculated i.p. with GFP-expressing ES2 VEGF$^{-/-}$ cells. At 7 days thereafter when tumors were palpable, mice were randomized into groups ($n = 6$ mice per group) and then administered sEVs of ES2 VEGF$^{+/+}$ cells in combination with either normal human IgG (negative control) or bevacizumab, or rVEGF$_{189}$ in combination with either normal human IgG or bevacizumab, three times a week for 2 weeks. In **a**, representative images of GFP-expressing tumors in the abdominal cavity. Arrows indicate tumors on the omentum. Scale bar = 10 mm. In **b**, immunofluorescence staining of CD31 (red) in sections of omental tumors (Ome T) adjacent to the pancreas (Panc). Scale bar = 100 μm. Amount of i.p. tumor burden (**c**), numbers of intratumoral CD31$^+$ cells (**d**) and volume of ascites (**e**) in each mouse in control (Cont) and bevacizumab (Bev) treatment groups. \*\*\*$P < 0.001$, \*\*\*\*$P < 0.0001$, by two-sided unpaired $t$-test. **f, g** Baseline plasma levels of total VEGF (**f**) and sEV-VEGF (**g**) in 17 patients with newly diagnosed metastatic renal cell carcinoma, who were treated presurgically with single-agent bevacizumab for 8 weeks and thereafter restaged. $P$-values were determined by Mann–Whitney $U$-test. Source data used for graphs in **c**–**g** can be found in Supplementary Data 7

similarly localized. We detected GROα and IL-8 in sEVs of some cancer cell lines and these ligands mostly localized on the surface (Fig. 2b). In contrast, most of the FGF-2 detected in sEVs was not surface-associated (Fig. 2b). This finding might be explained by the non-classical secretion of FGF-2. Whereas many other growth factors undergo endoplasmic reticulum-to-Golgi trafficking, FGF-2 lacks a signal peptide and is instead recruited to the inner leaflet of the plasma membrane[36]. Once there, FGF-2 undergoes oligomerization that in turn causes the formation of membrane pores, enabling FGF-2 to cross the membrane[36]. FGF-2 is also released from cells in ectosomes that bud from the plasma membrane and range from 100 to 1000 nm in diameter[37]. It is possible that our sEV preparations not only included exosomes but also small ectosomes. The formation of ectosomes through outward budding of the plasma membrane could explain why FGF-2 is encapsulated in these EVs rather than located on the EV surface.

Although functions of sEV cargo have been increasingly studied, the physiological relevance of levels of many of these constituents is unclear. sEV-VEGF constituted one-third of the total VEGF secreted by cancer cells that we analyzed (Fig. 3f) and up to one-third of the total VEGF in ascites of mice and women with ovarian cancer (Table 1). Furthermore, the analysis of xenograft models indicated that almost all sEV-VEGF was tumor-derived (Supplementary Fig. 9b). Although several growth factors have been detected in cancer-derived sEVs[8], it has been unclear as to whether vesicular localization alters properties of the ligands. VEGF has a short half-life, as is the case for most growth factors[38]. Notably, we found that interaction of VEGF$_{189}$ with the surface of sEVs profoundly increases ligand half-life (Fig. 6h). It has been thought that VEGF$_{189}$, by virtue of being membrane/matrix-bound, acts locally whereas VEGF$_{121}$, by virtue being freely secreted, mediates long-range signaling[39]. Our findings that sEV-VEGF predominantly comprises VEGF$_{189}$, is signaling-competent, highly stable, and present in the peripheral circulation of tumor-bearing mice and cancer patients collectively support the possibility that VEGF$_{189}$, through being conveyed on secreted sEVs, also mediates long-range signaling. Additional long-range signaling, mediated by sEV-VEGF, might provide a strong advantage to tumors and particularly for metastasis.

Another significant outcome of our findings that cancer cell-derived sEVs contain heparin-bound, signaling-competent VEGF is the impact on responsiveness of tumors to bevacizumab. Bevacizumab was initially approved for treatment of metastatic colorectal cancer and subsequently approved for several other solid tumors including ovarian cancer and metastatic renal cell carcinoma[24,28]. However, the clinical benefit of bevacizumab has not been as promising as first expected and its approval for treatment of metastatic breast cancer was withdrawn due to modest benefits and significant adverse effects[40]. Resistance of tumors to anti-VEGF therapy has been attributed to several mechanisms that are independent of VEGF signaling. These include the utilization by tumors of existing vasculature[41] and of alternative angiogenic pathways such as those mediated by Bv8, which is secreted by infiltrating myeloid cells[42] and by adipose tissue-derived IL-6 and FGF-2[43]. Resistance to anti-VEGF therapy has also been attributed to hypoxia-triggered metabolic reprogramming and increased uptake of free fatty acid that fuels tumor growth[44]. Poor outcomes can also stem from effects of anti-VEGF therapy on the endocrine system[45]. Our findings that sEV-VEGF is not neutralized by bevacizumab implicate that resistance to bevacizumab might also stem in part from the failure of this agent to recognize its target molecule. Bevacizumab has been thought to neutralize all isoforms of VEGF, but prior studies have largely focused on characterizing the binding of this agent to soluble VEGF. Several residues in the β5-sheet, β5-β6 loop, and β6-sheet of VEGF are critical for forming a high-affinity complex

with bevacizumab[25]. Intriguingly, interaction of VEGF with HMW heparin substantially decreases the β-sheet content of VEGF and increases its α-helix content[46]. We found that binding of bevacizumab to VEGF$_{189}$ was substantially reduced when VEGF$_{189}$ was engaged with HMW heparin (Fig. 7b). The inability of bevacizumab to neutralize sEV-VEGF might therefore stem, at least in part, from conformational change in this ligand that is induced through its interaction with heparin.

A limitation of bevacizumab has been the lack of robust biomarkers that can predict clinical response[28,47]. Whereas some studies have found that baseline plasma levels of total VEGF correlate with outcomes following bevacizumab treatment[48,49], there are several other reports that baseline levels of total VEGF are not predictive of bevacizumab treatment benefit[47,50,51]. In an independent cohort of patients with renal cell carcinoma who received bevacizumab monotherapy, we found no significant difference in baseline levels of total VEGF between patients who had progressing disease and those who had stable or regressing disease (Fig. 8f). In contrast, baseline levels of sEV-VEGF were ~5-fold higher in patients with progressing disease than in those with stable or regressing disease (Fig. 8g). Intriguingly, IEV-associated VEGF$_{90K}$ has also been found to be bevacizumab insensitive[32]. However, it is as yet unclear whether this form of VEGF is present in body fluids of cancer patients and to what extent it contributes to the total circulating VEGF.

In summary, our study shows that VEGF$_{189}$, but not two other major isoforms of VEGF, is selectively enriched in cancer cell-derived sEVs, associates with the surface of sEVs via heparin-binding, and can be delivered in signaling-competent form by sEVs to endothelial cells independently of EV uptake. Our study also shows that interaction of VEGF$_{189}$ with the surface of sEVs profoundly increases ligand half-life, and that sEV-VEGF is not neutralized by bevacizumab. Our findings implicate that resistance of tumors to bevacizumab might stem in part from the ability of cancer cell-derived sEVs to deliver biologically active VEGF to recipient cells without being recognized by the neutralizing agent. Although validation in large independent cohorts is needed, our findings suggest that baseline levels of sEV-VEGF might be more informative for bevacizumab treatment benefit than levels of total VEGF. Furthermore, our findings that the activity of sEV-VEGF can be blocked by VEGFR tyrosine kinase inhibitors or VEGFR2-neutralizing Ab raise the possibility that treatment with these inhibitors rather than bevacizumab might be beneficial for patients who have elevated sEV-VEGF levels.

## Methods
**Reagents.** Sources of Ab were as follows: anti-VEGF (for detection: R&D Systems MAB2931, Thermo Fisher Scientific P802, Abcam ab52917, ab183100; for capture: R&D Systems MAB293; for neutralization (bevacizumab): Genentech); anti-VEGF-phycoerythrin (PE) (Abcam ab209439); anti-VEGFR2 (for detection: Cell Signaling Technology 9698; for neutralization: R&D Systems MAB3572); anti-phospho-VEGFR2 (Tyr1175); anti-calnexin (Cell Signaling Technology 3770; 2679); anti-CD63 (for flow cytometry: BD Biosciences 556019; for immunogold labeling: System Biosciences EXOAB-CD63A-1); anti-CD63-PE/Cy7 (BioLegend 353010); anti-flotillin-1, anti-HSP70 (BD Biosciences 610820, 610607); anti-TSG101, anti-α-actinin-4, anti-CD31, anti-green fluorescent protein (GFP) (Abcam ab125011, ab108198, ab28364, and ab5450); anti-HSP90B1 (Enzo Life Sciences ADI-SPA-850); anti-bevacizumab (Abnova mab11128); anti-actin, anti-human IgG (Fc specific)-horseradish peroxidase (HRP) (Sigma-Aldrich A1978, A0170); normal human IgG (Innovative Research IR-HU-GF-ED); normal mouse IgG (Thermo Fisher Scientific 10400C); 10 nm anti-rabbit IgG gold conjugate (Electron Microscopy Sciences 25109); other secondary Ab (BioLegend 409312, 400126; BD Biosciences 340272; Abcam ab97263, ab97198, ab97223; Jackson ImmunoResearch 115-035-166, 111-035-144; Santa Cruz Biotechnology sc-3739). Concentrations of Ab used are specified in the assays described below. Enzyme-linked immunosorbent assay (ELISA) kits were as follows: human VEGF, GROα, IL-8, and FGF-2, mouse VEGF (R&D Systems DVE00, DGR00B, D8000C, DFB50, MMV00); human CD63, TSG101 (LSBio LS-F11093, LS-F8581). VEGFR1/R2-Fc (Aflibercept, Regeneron Pharmaceuticals) was provided by D.M. Brown (Houston Methodist Hospital). Other reagents were as follows: recombinant VEGF$_{121}$ (Shenandoah Biotechnology);

recombinant VEGF$_{165}$ (BioLegend); recombinant VEGF$_{189}$, dynasore (R&D Systems); axitinib, vandetanib (Cell Signaling Technology); dimethyl sulfoxide (DMSO), chlorpromazine, heparinase I and III blend from *Flavobacterium heparinum*, chondroitinase ABC from *Proteus vulgaris*, 4′,6-diamidino-2-phenylindole (DAPI) (Sigma-Aldrich); fractionated heparan sulfate polymer (molecular weight ~40 kDa) (Amsbio).

**Cell culture**. Culture media was purchased from Corning. HUVEC were purchased from American Type Culture Collection (ATCC) and cultured in Medium 199 supplemented with 20% fetal bovine serum (FBS), endothelial cell growth supplement (50 µg/mL) (Millipore), L-glutamine, and penicillin–streptomycin on plates coated with 0.1% gelatin (Sigma-Aldrich). Parental ES2, HCT116, and 786-0 cell lines were purchased from ATCC, confirmed to be free of mycoplasma contamination, and authenticated by short tandem repeat analysis. Cancer cell lines were cultured in McCoy's 5A medium (ES2) or Dulbecco's modified Eagle's medium (DMEM) (HCT116, 786-0) supplemented with 10% FBS and penicillin–streptomycin. HCT116 VEGF$^{-/-}$ cells were generated in a previous study by targeting exon 2 of the *VEGFA* gene for disruption with an adeno-associated virus knockout construct[15]. GFP-expressing ES2 VEGF$^{-/-}$ cells were generated in this study by CRISPR/Cas9 gene editing using the human *VEGFA* gene knockout kit (Origene). Briefly, ES2 cells were co-transfected with *VEGFA* sgRNA plasmid and donor vector that contains predesigned homologous sequences flanking the GFP and puromycin resistance selection cassette. Individual clones, derived from single GFP-expressing cells, were selected with puromycin (0.5 µg/mL) and screened by ELISA to confirm *VEGFA* gene knockout. CHO-K1 and pgsD-677 cell lines were purchased from ATCC and cultured in Ham's F-12 medium supplemented with 10% FBS and penicillin–streptomycin. In VEGF reconstitution experiments, 293T cells (purchased from ATCC) were transfected with lentiviral constructs encoding VEGF$_{121}$, VEGF$_{165}$, and VEGF$_{189}$ (Genecopoeia) by using Lipofectamine 3000 (Thermo Fisher Scientific). At 2 days thereafter, supernatants containing lentiviral particles were collected and used to infect ES2 VEGF$^{-/-}$, HCT116 VEGF$^{-/-}$, CHO-K1, and pgsD-677 cells.

**Clinical specimens**. Studies using human tissue specimens were approved by the Institutional Research Board of the University of Texas MD Anderson Cancer Center (UTMDACC) and the Institutional Research Board of the University of Chicago. Full informed consent was obtained from all human subjects. All specimens used in this study were residual and not necessary for diagnosis. To analyze cellular VEGF and sEV-VEGF, specimens of tumor tissue and ascites of women with ovarian carcinoma were obtained from the Ovarian Cancer Tumor Bank at the University of Chicago. Serum and plasma samples of individuals with colorectal and renal cell carcinoma were obtained from the National Cancer Institute-supported Cooperative Human Tissue Network. Residual blood samples from healthy adult donors were provided by the UTMDACC Blood Bank. To analyze relationships between VEGF levels and outcomes following bevacizumab treatment, residual plasma samples from study NCT00113217[29] were obtained from the Eckstein Tissue Acquisition Laboratory at UTMDACC. These samples were collected from patients who were enrolled in study NCT00113217 at UTMDACC between March 2005 and March 2008. Collection, processing, and analysis of plasma samples were performed at UTMDACC. Of the 27 patients with newly diagnosed metastatic renal cell carcinoma, who were treated presurgically with single-agent bevacizumab and thereafter restaged in study NCT00113217[29], residual baseline plasma samples were available for 17 evaluable patients with Stage IV disease. To eliminate bias, all assays of VEGF levels in plasma samples were performed blinded to clinical data.

**Animal studies**. Animal studies were conducted in compliance with protocols approved by the UTMDACC Institutional Animal Care and Use Committee. Four-week-old female nude mice (purchased from UTMDACC animal facility) were used to propagate xenografts. To evaluate VEGF levels in ascites, mice were inoculated i.p. with $1 \times 10^6$ ES2 VEGF$^{+/+}$ cells and were killed by CO$_2$ asphyxiation upon formation of morbid ascites (median survival time of 3 weeks). To evaluate the effects of sEVs, mice were inoculated i.p. with $2 \times 10^6$ GFP-expressing ES2 VEGF$^{-/-}$ cells. At 7 days thereafter when tumors were palpable, mice were randomized into groups ($n = 6$ mice per group) and administered either phosphate-buffered saline (PBS), sEVs of ES2 VEGF$^{+/+}$ cells or of ES2 VEGF$^{-/-}$ cells (500 µg per animal) or recombinant human VEGF (5 ng per animal), alone or in combination with normal human IgG (5 mg/kg) or bevacizumab (5 mg/kg). Mice were treated with these agents i.p. three times a week for 2 weeks and thereafter killed. A 5 mg/kg dose of bevacizumab has been used by other investigators to treat mice with xenografts derived from ES2 VEGF$^{+/+}$ cells[26]. The six doses of sEVs at 500 µg per dose was based on the average volume of ascites and concentration of tumor-derived sEV-VEGF in ascites that forms in mice with i.p. xenografts derived from ES2 VEGF$^{+/+}$ cells (Supplementary Fig. 9a, b). Following killing, volumes of ascites were measured. GFP-expressing tumors were visualized under a Leica MZMLIII stereomicroscope equipped with a GFP filter set and digital camera. Images were captured by using Picture Frame software (Optronics). Tumor burden was quantified by measuring areas of fluorescence signals within the abdominal cavity in captured images by using Image Pro Plus software (Media

Cybernetics) as performed in our previous work[52]. To visualize intratumoral blood vessels, sections of omental tumor tissues, frozen in Optimal Cutting Temperature compound (Thermo Fisher Scientific), were stained with Ab to GFP (1:500 dilution) and CD31 (1:50 dilution), and with DAPI, and viewed under a Nikon 80i fluorescence microscope. Numbers of intratumoral CD31$^+$ cells were counted in five random 100× fields per section and an average score was determined for each mouse. To evaluate circulating VEGF levels, mice were inoculated s.c. with $5 \times 10^6$ ES2 VEGF$^{+/+}$ cells or HCT116 VEGF$^{+/+}$ cells. At 7 days thereafter when tumors were palpable, blood samples were collected retro-orbitally. Mice were then randomized into groups ($n = 3$–4 mice per group) and administered either normal human IgG (5 mg/kg) or bevacizumab (5 mg/kg) i.p. three times per week followed by retro-orbital blood collection. Volumes of s.c. tumors were calculated from two perpendicular measurements of tumor diameters taken using calipers. Pre- and post-treatment plasma samples were pre-cleared with Protein G Sepharose 4 Fast Flow (GE Healthcare) at 4 °C for 16 h to remove Ab-bound VEGF and thereafter were assayed for VEGF by ELISA.

**Isolation of sEVs**. In all experiments, sEVs were isolated from conditioned media and body fluids (with the exception of plasma samples in Fig. 8g) by the following procedure. Conditioned media was prepared by culturing cancer cells in media containing 2% FBS for 48 h. For each batch preparation of sEVs, a total of 360 mL of conditioned media collected from 20 × 150 mm dishes of cells at 90% confluence was used. For each purification of sEVs from biological fluid, a 1 mL sample of fluid was used. Conditioned media and biological fluids were centrifuged at 2,400 × g at 4 °C for 10 min to remove intact cells and cell debris. Thereafter, supernatants were filtered through a 0.2 µm pore size filter to exclude particles of > 200 nm in diameter and then concentrated using a Centricon® Plus-70 centrifugal filter unit with a 100 kDa nominal molecular weight limit (Millipore) to exclude soluble proteins of < 100 kDa in size. To prepare the discontinuous iodixanol gradient, solutions of iodixanol were prepared by diluting a stock solution of OptiPrep™ (60% (w/v) aqueous iodixanol, Axis-Shield PoC) in buffer containing 0.25 M sucrose, 10 mM Tris-HCl (pH 7.4), and 1 mM EDTA. Concentrated conditioned media and biological fluids were mixed with 1.5 mL of Optiprep™ stock solution on the bottom of a 14 × 95 mm polyallomer ultracentrifuge tube (Beckman Coulter). The gradient was formed by stepwise layering of 3.0 mL of 40% (w/v) iodixanol solution, 2.5 mL each of 20% (w/v) and 10% (w/v) iodixanol solutions, and 2.0 mL of 5% (w/v) iodixanol solution. Centrifugation was performed at 200,000 × g at 4 °C for 18 h. Ten gradient fractions of 1.0 mL were collected from the top to bottom. The density of each fraction was determined from absorbance readings at 244 nm using a standard curve generated from serial dilutions of iodixanol solution[53]. Individual fractions were washed with PBS, concentrated by using Centricon® filter units, and suspended in PBS for further analysis. As volumes of plasma samples of bevacizumab-treated patients were small (200 µL), sEVs were isolated from these samples (Fig. 8g) by using ExoQuick reagent (System Biosciences). Briefly, plasma samples were centrifuged at 3000 × g for 15 min to remove cells and debris. Samples (100 µL) were then diluted with the addition of 400 µL PBS and 120 µL precipitation solution, vortexed and incubated at 4 °C for 16 h. Thereafter, sEVs were precipitated by centrifugation at 1500 × g for 30 min and suspended in PBS for further analysis.

**Depletion of sEVs**. Samples of plasma and ascites (diluted 1:10 in PBS) and conditioned media were centrifuged at 2400 × g at 4 °C for 10 min to remove intact cells and cell debris. Thereafter, supernatants were filtered through a 0.2 µm pore size filter to remove larger particles. A 300 µL aliquot of filtrate was retained for analysis of VEGF. The remainder of the filtrate was centrifuged at 100,000 × g at 4 °C for 18 h to remove sEVs. Thereafter, the supernatant (comprising of sEV-depleted sample) was assayed for VEGF.

**Particle size analysis**. Particle size distribution of purified sEVs was evaluated by nanoparticle tracking analysis using a Nanosight LM10 (Malvern) by Alpha Nano Tech LLC. For each batch purification of sEVs, an average of $2 \times 10^9$ vesicles was isolated. Ten replicate measurements were made for an individual batch of sEVs.

**Immunogold labeling and transmission electron microscopy**. For immunogold labeling, carbon-coated, formvar-coated nickel grids (200 mesh) were treated with poly-L-lysine for 30 min. Samples were then loaded and allowed to absorb for 1 h. Grids were placed into buffer containing 2% bovine serum albumin (BSA) and 0.1% saponin for 20 min, and then placed into CD63 Ab (1:5 dilution) or VEGF Ab (1:10 dilution) at 4 °C for 16 h. Control grids were incubated without primary Ab. Thereafter, grids were rinsed with PBS and floated on drops of anti-rabbit IgG labeled with 10 nm gold particles (1:20 dilution) for 2 h at room temperature (RT). Following washing with PBS, grids were placed in 1% glutaraldehyde for 5 min and washed in H$_2$O. Grids were stained for contrast for 1 min with 1% uranyl acetate and allowed to dry. Samples were evaluated under a JEM 1010 transmission electron microscope (JEOL USA, Inc.) at an accelerating voltage of 80 Kv. Images were obtained by using the AMT Imaging System (Advanced Microscopy Techniques, Corp.).

**Fluorescence labeling of sEVs.** sEVs were incubated with PKH67 green or PKH26 red fluorescent linker dye (Sigma-Aldrich) (1:500 dilution) in a final volume of 100 μL at RT for 10 min. Reactions were stopped by adding 100 μL of 5% BSA in PBS. sEVs were precipitated using Exoquick-TC solution (System Biosciences) and resuspended in PBS. As described further below, sEVs were also labeled with exo-FITC dye (Systems Biosciences).

**Analysis of sEV uptake.** HUVEC ($10^5$ cells) were incubated with PKH26-labeled sEVs (100 μg/mL) at 37 °C for times indicated in the legends. As a negative control, HUVEC were incubated with PKH26 linker dye precipitated with Exoquick-TC solution without sEVs to exclude the presence of aggregates of linker dye. Uptake of sEVs in HUVEC was visualized under a Nikon 80i fluorescence microscope attached to a digital camera. Images were captured by using NIS Element BR 4 software (Nikon). Uptake of sEVs was also evaluated by measuring PKH26 fluorescence intensity in the gated population of viable HUVEC by using a FACSCalibur™ cytometer equipped with CellQuest™ Pro software (BD Biosciences). A minimum of 10,000 gated events was analyzed for each sample. Three independent experiments were performed for each assay.

**Treatment of cells with inhibitors.** Uptake of sEVs was blocked by treating HUVEC with chlorpromazine (15 μM) or dynasore (50 μM) for 30 min prior to addition of sEVs. As a negative control, HUVEC were pretreated with DMSO solvent (0.1%). VEGFR signaling was blocked by treating HUVEC with axitinib (1 nM) or vandetanib (1 μM) for 2 h or with neutralizing Ab to VEGFR2 (250 ng/mL) for 1 h prior to addition of sEVs or recombinant VEGF. In other experiments, sEVs (100 μg/mL) or recombinant VEGF (1 ng/mL) were incubated with either normal human IgG, bevacizumab, or VEGFR1/R2-Fc (50 ng/mL) for 16 h. Thereafter, mixtures were added to HUVEC cultures. To detect intracellular VEGF, cancer cells where indicated were incubated at 37 °C for 6 h with brefeldin A (BioLegend) (5 μg/mL) to block protein secretion and then collected.

**Tube formation assay.** Wells of 96-well plates were coated with growth factor-reduced Matrigel (BD Biosciences) (50 μL/well) and incubated at 37 °C for 30 min. HUVEC ($10^4$ cells/well) were then seeded in FBS-free Medium 199 without or with addition of sEVs (100 μg/mL) or recombinant VEGF (1 ng/mL), or seeded in whole or sEV-depleted conditioned media. Following incubation at 37 °C for 4 h, capillary tube structures were visualized under a Nikon TS100 light microscope attached to a digital camera. In each experiment, images of two to three random 100× fields of each well were captured by using NIS Element BR 4 software (Nikon). The number of tubes in each field was quantified by using NIH ImageJ software with Angiogenesis Analyzer plugin and an average calculated for each well. Three to four independent experiments were performed for each assay, where each experiment used a different batch of sEVs.

**Migration assay.** sEVs (100 μg/mL) or recombinant VEGF (1 ng/mL) were suspended in Medium 199 and added to the lower chamber of 24-well transwell chambers (Corning). HUVEC ($10^5$ cells/well) were seeded in the upper chamber and incubated at 37 °C for 5 h. Migrating cells were then fixed, stained with crystal violet solution (Sigma-Aldrich), and visualized under a Nikon TS100 light microscope attached to a digital camera. In each experiment, images of two to three random 100× fields of each well were captured by using NIS Element BR 4 software. The number of migrating cells in each field was manually counted and an average calculated for each well. Four independent experiments were performed for each assay, where each experiment used a different batch of sEVs.

**Immunoblot analysis.** Extracts were prepared by lysing whole cells and sEVs in M-PER buffer (Thermo Fisher Scientific). Protein concentrations of lysates were determined by Bradford assay (BioRad). Lysates were separated by SDS-polyacrylamide gel electrophoresis under non-reducing conditions to detect VEGF and under reducing conditions to detect other proteins, and then transferred to polyvinylidene difluoride membrane (GE Healthcare). Membranes were blocked with 5% non-fat milk in Tris-buffered saline with 0.1% Tween-20 (TBS-T), incubated with primary Ab at 4 °C for 16 h, and then washed with TBS-T buffer. Primary Ab were used at the following dilutions: 1:1000 (for Ab to TSG101, flotillin, α-actinin-4, HSP90B1, calnexin, phospho-VEGFR2, VEGFR2, VEGF); 1:2000 (for Ab to HSP70); 1:5000 (for Ab to actin). Thereafter, membranes were incubated with HRP-conjugated secondary Ab (1:5000 dilution), washed, and visualized with ECL detection reagent (Millipore). Immunoblot data were verified in three independent experiments.

**Detection of angiogenic factors by Ab array and ELISA.** Angiogenesis-related proteins were detected in sEVs by using the Human Angiogenesis Antibody Array (Abcam). sEVs were lysed in buffer provided by the manufacturer. Membranes were incubated with sEV lysate (100 μg) at 4 °C for 16 h, and then incubated with Ab cocktail and visualized according to manufacturer's instructions. Levels of VEGF, GROα, IL-8, FGF-2, CD63, and TSG101 in sEVs were quantified by ELISA as follows. Purified sEVs were divided into identical aliquots. One set of aliquots was lysed in passive lysis buffer (Promega). Lysates of sEVs were assayed by ELISA

to determine the total content of a given protein in sEVs, expressed relative to total protein content in sEVs. Other aliquots of sEVs were left intact and assayed by ELISA to determine the amount of a given protein on the surface of sEVs. ELISA data were measured by using a ELx800 microplate reader equipped with Gen 5 software (BioTek). Three independent experiments were performed for each assay.

**Detection of sEV-associated proteins by flow cytometry.** To detect sEV-associated proteins by using Ab-coupled microbeads, acquisition and analysis of flow cytometry data were performed using a FACSCalibur™ cytometer equipped with CellQuest™ Pro software (BD Biosciences). Initially, Exo-Flow microbeads ($6.4 \times 10^5$ beads of 9.1 μm diameter) (System Biosciences) were coupled with the appropriate secondary Ab (1 μg) and then with the desired detection Ab (i.e., 1 μg for anti-VEGF, anti-CD63, anti-TSG101, bevacizumab, VEGFR1/R2-Fc) following the manufacturer's instructions. Thereafter, microbeads were incubated with sEVs (100 μg, ~$2 \times 10^8$ vesicles) at 4 °C for 16 h with rotation, washed with 1% BSA in PBS, stained with exo-FITC dye, and then acquired. Binding of Ab to protein on the surface of sEVs was evaluated by analyzing exo-FITC fluorescence in the gated population of singlet microbeads. A minimum of 10,000 gated events was analyzed for each sample. To confirm that bevacizumab was coupled to microbeads, microbeads were stained with Ab to bevacizumab (1:100 dilution) at 4 °C for 30 min, washed with 1% BSA in PBS, and then stained with PerCP-conjugated anti-mouse IgG (1:100 dilution) at 4 °C for 30 min. Following washing, microbeads were acquired and PerCP fluorescence analyzed in the gated population of singlet microbeads. Where indicated in the text, sEV-associated surface proteins were assayed following enzymatic treatment. PKH67-labeled sEVs (100 μg) were suspended in digestion buffer (DMEM supplemented with 0.5% BSA and 20 mM HEPES-HCl (pH 7.4)) and incubated with heparinase I and III blend (1 mU/mL) or with chondroitinase ABC (50 mU/mL) at 37 °C for 3 h. Fresh enzyme was then added and samples incubated for a further 16 h. sEVs were then extensively washed with PBS, concentrated by using Centricon® filter units, and then incubated with Ab-coupled microbeads. Binding was evaluated by analyzing PKH67 fluorescence in the gated population of singlet microbeads. Three independent experiments were performed for each assay. Contour plots were generated by using CellQuest™ Pro software. To detect sEV-associated proteins by direct staining of sEVs, acquisition and analysis of flow cytometry data were performed using a BD FACSCanto II cytometer equipped with FACS Diva software (BD Biosciences). Settings were optimized by using bead calibration kits (50 nm, 100 nm, 200 nm, and 500 nm diameter beads) purchased from Bangs Laboratories. A 100 μL aliquot of sEV sample (100 μg, ~$2 \times 10^8$ vesicles) was incubated at RT for 30 min with PE-conjugated VEGF Ab or isotype control (1:100 dilution) in combination with PE/Cy7-conjugated CD63 Ab or isotype control (1:100 dilution). Following incubation, samples were diluted to final volume of 500 μL in PBS and acquired. PE and PE/Cy7 fluorescence was analyzed in the gated population of sEVs. A minimum of 10,000 gated events was analyzed for each sample. Contour plots were generated by using FlowJo v10.6.0 software (FlowJo LLC).

**Detection of cellular proteins by flow cytometry.** For detecting intracellular VEGF, brefeldin A-treated cancer cells were fixed with 1% paraformaldehyde at 4 °C for 20 min and then permeabilized in 0.1% saponin at RT for 15 min. Following washing with 1% BSA in PBS, cells were incubated with bevacizumab or VEGFR1/R2-Fc (25 μg/mL) at 4 °C for 30 min and then washed. Cells were stained with PerCP-conjugated anti-human IgG (Fc specific) (1:100 dilution), then washed and fixed. Cells were acquired by a FACSCalibur™ flow cytometer and staining analyzed by CellQuest™ Pro software. A minimum of 10,000 gated events was analyzed for each sample.

**Analysis of VEGF stability.** Stability of VEGF protein was assayed in healthy adult donor plasma as follows. Recombinant VEGF (20 ng) or sEVs (2 mg) were added to 400 μL of 10% plasma and placed at 37 °C. At 0, 0.5, 1, 3, 6, 9, 12, 16, and 24 h thereafter, aliquots of 40 μL were removed and immediately frozen. Following collection of all samples, thawed samples were diluted and then assayed for VEGF content by ELISA.

**Detection of heparin-bound VEGF.** Wells of 96-well High Bind microplates (Corning) were coated with fractionated heparan sulfate polymer (5 μg) or with VEGF capture Ab (100 ng) at RT for 16 h. Wells were then washed three times with wash buffer (R&D Systems), incubated with recombinant $VEGF_{189}$ (2 ng) for 2 h, and washed again. Thereafter, bevacizumab or VEGFR1/R2-Fc (100 ng in a reaction volume of 100 μL) were added to wells that contained $VEGF_{189}$ bound to heparan sulfate polymer or to VEGF capture Ab, and incubated for 2 h at RT. Wells were then washed three times. Binding of bevacizumab and VEGFR1/R2-Fc was detected by incubation with HRP-conjugated goat anti-human IgG (Fc-specific) (1:5000 dilution) for 1 h at RT, followed by washing and addition of HRP substrate solution (R&D Systems). Absorbance was read at 450 nm. The reading of wells that contained heparin-bound $VEGF_{189}$ was normalized to the reading of wells that contained VEGF bound to VEGF capture Ab.

**Quantification of unbound bevacizumab and VEGFR1/R2-Fc.** Bevacizumab (5 ng) or VEGFR1/R2-Fc (5 ng) were incubated at 4 °C for 16 h with a range of amounts of sEVs and recombinant VEGF proteins indicated in the legends in a reaction volume of 100 μL. In other experiments, 100 μL of conditioned media was pre-cleared with 50 μL of Protein G Sepharose 4 Fast Flow at RT for 2 h to remove Ab, and then incubated with the addition of bevacizumab (2.5 ng) at 4 °C for 16 h. Following incubation, concentrations of VEGF-unbound bevacizumab and VEGF-unbound VEGFR1/R2-Fc in reaction samples and conditioned media were assayed by ELISA as follows. Wells of 96-well High Bind microplates were coated with recombinant VEGF (100 ng) at RT for 16 h. After washing three times with wash buffer (R&D Systems), wells were blocked with 10% goat serum for 1 h and washed again. Thereafter, reaction samples, conditioned media, and bevacizumab or VEGFR1/R2-Fc controls (100 μL) were added to wells. Standard controls of bevacizumab and VEGFR1/R2-Fc were prepared at concentrations ranging from 0.78 to 50 ng/mL. VEGF-unbound bevacizumab and VEGF-unbound VEGFR1/R2-Fc (i.e., captured by precoated recombinant VEGF) were detected by incubation with HRP-conjugated goat anti-human IgG (Fc specific) (1:5000 dilution) for 1 h at RT, followed by washing and addition of HRP substrate solution. Absorbance was read at 450 nm. The concentration of VEGF-unbound bevacizumab or VEGF-unbound VEGFR1/R2-Fc in each reaction was calculated from the relevant standard curve.

**Statistics and reproducibility.** Statistical analysis was performed by using STATISTICA13.1 (StatSoft, Inc.) and Prism8 software (GraphPad Software, Inc.). For each type of in vitro assay, two to six independent experiments (i.e., using independent samples) were performed to confirm reproducibility. Based on the variance of i.p. xenograft growth that was observed in control mice in preliminary studies, power calculations indicated the use of $n = 6$ mice per group to detect a difference of > 39% in i.p. tumor burden, numbers of intratumoral endothelial cells, and ascites volume between groups in a two-sided test at a significance of $P < 0.05$ and with 80% probability. For studies of clinical specimens, sample size was limited by availability of archived specimens. Normality of data distribution in groups was assessed by Shapiro–Wilk test. Significance of data in in vitro and in vivo assays was assessed by one-way or two-way analysis of variance with Bonferroni's corrections for multiple comparisons, or by unpaired or paired two-tailed Student's $t$-test. Significance of data between patient groups was assessed by Mann–Whitney $U$-test, as data in these groups was non-normally distributed. $P$-values of < 0.05 were considered significant.

**Reporting summary.** Further information on research design is available in the Nature Research Reporting Summary linked to this article.

## Data availability

All relevant data generated in this study are included in this article, Supplementary Information, and additional Supplementary Files.

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

## Acknowledgements

We thank J. Yao, J. V. Vykoukal, and H. Akasaka (UTMDACC) for helpful discussions and the animal, electron microscopy, flow cytometry, histopathology, and characterized cell line core facilities supported by UTMDACC Core Grant (NCI CA16672) for technical assistance. We thank staff of the Ovarian Cancer Tumor Bank (University of Chicago) and Eckstein Tissue Acquisition Laboratory (UTMDACC) for retrieving clinical specimens. This work was supported by National Institutes of Health grants CA207034 (to H.N.), CA217931 (to H.N.), and CA169604 (to E.L.), and an Ovarian Cancer Research Fund Ann and Sol Schreiber Mentored Investigator Award (to S.Y.K.).

## Author contributions

S.Y.K. and H.N. developed the original hypothesis, conceived the study, and designed experiments. S.Y.K., W.L. and H.N. performed experiments and analyzed data. H.A.K., E. L. and E.J. provided clinical specimens. L.M.E. provided bevacizumab. L.H.D. provided HCT116 cell lines. S.Y.K. and H.N. wrote and edited the manuscript. W.L., H.A.K., L.M.E, E.J. and E.L. edited the manuscript. H.N. supervised the study.

## Competing interests

The authors declare no competing interests.
