## [Peer Review File · Communications Biology]

Reviewers' comments:

Reviewer #1 (Remarks to the Author):

In this manuscript, Ko et. al characterize the association of VEGF with sEVs and test the functional importance of this in cancer. They present interesting data showing a VEGF specific isoform association with sEVs that impacts VEGF stability and resistance to the drug bevacizumab. Overall, the experiments are clear and the data is well-presented. Additionally, this description of VEGF and sEVs adds to our knowledge of selective protein association with EVs and offers some new insight into distinctions between EV-associated and soluble proteins. Because VEGF and EVs are both areas of intense study in cancer, these findings are likely to be of interest to a wide audience. That said, there are a couple of major conceptual issues the authors should address experimentally or offer some discussion for to better place their findings into proper context.

- Although the authors do a nice job of validating their sEV preparations from cell lines following centrifugation, filtration, and density gradient ultracentrifugation in Fig 1 and Suppl Fig 1 and 2, some issues remain regarding their EV isolation and characterization. First, for human samples and experiments with PKH26 labeling, they have isolated sEVs using the Exoquick kit without showing any validation of those particular preparations. Because of the ability of Exoquick to purify non-EV impurities due to its polymer-based precipitation, Exoquick is not a commonly accepted method for sEV purification. Thus, it will be necessary for the authors to show better characterization of these particular sEV preparations, confirming they lack non-EV contaminants, or repeat these experiments using more conventional approaches, namely ultracentrifugation. Second, although the authors confirm that VEGF is found in EV fractions using density gradient centrifugation, they describe this as being a marked enrichment in these fractions, when in fact VEGF is also equally abundant in other fractions (Suppl Fig 6 purple and gray bars). Similarly, the authors also show that there is abundant non-sEV associated VEGF; therefore, effort should be invested into verifying that VEGF is actively sorted into EVs and is not sticking to them as a result of being co-secreted. While the authors provide evidence that VEGF is sticking to the surface of sEVs via heparin binding, this could be due to active packaging of VEGF into sEVs or, alternatively, binding of VEGF to heparin-containing sEVs in conditioned medium or circulation following secretion. This post-secretion interaction would still allow for EV-associated VEGF to be identified in EV fractions by density gradient separation. Furthermore, this interaction would be affected by perturbations that alter levels of heparin on sEV surfaces, so the experiments in Fig 5 are not necessarily specific to EV-sorted VEGF. Regardless of the route by which VEGF becomes associated with sEVs, this sEV-bound VEGF may still be important, as the authors go on to assess, but this distinction should be clarified or these possibilities discussed.

- It is interesting that VEGF bound to heparin on sEVs is particularly stable and also resistant to bevacizumab. However, based on the data presented, the physiological relevance of this finding is unclear. First, although the authors show in Fig 5e that sEV-bound VEGF189 is more stable compared to free, recombinant VEGF189, this particular experiment does not accurately model a physiological situation in which it would be expected that VEGF or sEVs carrying VEGF are secreted on a more continuous basis. Thus, any VEGF that is turned over or degraded would be actively replaced, bringing into question the need for enhanced stability of VEGF via sEV secretion. Moreover, the authors have calculated in Table I that the majority of VEGF is not sEV-associated, suggesting that there is little necessity for this enhanced stability of VEGF through sEV secretion since most of the secreted VEGF is soluble. Along these lines, while VEGF189 was found to be particularly sEV-associated compared to other isoforms, the other isoforms are still secreted in soluble form (more so than 189 by HCT116 cells as shown in 5a). Finally, the authors suggest VEGF secreted with sEVs signals locally to endothelial cells, so again, the need for and impact of enhanced stability is unclear given the short-range paracrine signaling within the tumor microenvironment. Overall, it seems a small amount of VEGF would be protected by being bound to the surface of sEVs so the relevance of this protective effect, its relative contribution to the tumor microenvironment, and how it may promote bevacizumab resistance

in a meaningful way is unclear. Indeed, treatment of tumor-bearing mice in the absence of exogenously administered exosomes with the drug is sufficient to limit tumor growth and tumor angiogenesis (Fig 7), suggesting this protection of VEGF by endogenously secreted EVs is not enough to confer bevacizumab resistance. Although the patient data in 7d seems supportive of the author's conclusions, the majority of patients with progressing disease have sEV-VEGF levels comparable to patients with stable or regressing disease.

Specific comments:

- In the introduction
 - o The authors state that EV classification is based on size. However, it is also based on the cellular origin of the vesicles. Although the authors conservatively, and appropriately, use the term sEV to describe the vesicles they are studying since they have distinguished exosomes versus small microvesicles, it would be a necessary for them to clarify for their readers that classification is not just based on size, if they indeed want to make this statement.
 - o The authors should also consider reframing the goal of their study, as there is already published work on how VEGF carried by EVs affects endothelial cells (e.g., see Zhao et. al *Biochem Biophys Res Commun* 2018, Maisto et. al *Cell Cycle* 2019). Thus, the impact of this present work will not be to understand how sEVs affect "stromal cells through mechanisms other than transferring their luminal cargo..." as stated at the end of the introduction. This will provide readers with a broader picture of the field and references to other papers of interest.
- The detailed characterization of the EVs the authors are studying is appreciated. In Fig 1b EM images, the authors should also provide a lower power or wider field image to give a sense for the proportion of vesicles that stain for CD63.
- In Fig 1, it is surprising the inhibition of endocytosis does not affect endothelial cell migration and tube formation, as endocytosis of receptors that regulate motility is usually key for proper function and signaling of these receptors during migratory processes. Although the authors mention in the discussion that recycling/trafficking/endocytosis of VEGFR may not be necessary in this context, other membrane receptors involved in motility do, so it would be expected that these would be affected by the inhibitors. Can the authors explain this?
- Fluorescence microscopy images of the cells should also be provided along with the flow cytometry data in Fig 1 g, h and Suppl Fig 3, 4 to show how the sEVs interact with HUVECs and the differences in uptake with the inhibitors.
- Based on Fig 2b, VEGF is described as being the most abundant angiogenic factor on sEV surfaces. However, when using immunoassays of any sort, it is not possible to comment on the relative abundance of different factors, as this will be dictated by the performance of the antibody being used and different antibodies need to be used for different proteins. This point is underscored by comparing 2b to 2a, where the intensity of VEGF is actually the lowest while it gives the strongest signal by ELISA in 2b; this disparate expression signal is also evident for other factors measured, such as IL8, which, compared to VEGF, clearly gives a stronger signal in a but gives lower pg/ml measurements in b. If mass spec proteomics is performed, then it is possible to compare the levels of different proteins.
- For Fig 2c, please provide validation of VEGF depletion by western in the cells and ELISA in the supernatant.
- For Fig 2d, as with EM in Fig 1, low power/wide field images are necessary to assess the prevalence of EVs with VEGF. Use of EVs from VEGF negative cells would be a good control for this EM too.
- For Fig 2 f and g, representative images of the quantification should be presented. For these experiments, as well as 2e, it would also be helpful to use EVs from the VEGF negative cells presented in 2c.
- For Fig 4a and 4b, the authors draw comparisons between the expression levels of the different isoforms in cells and the isoform levels between cells and EVs. They state that 165 and 121 are the predominant forms in cells, but based on the blot in 4a, all isoforms seem equally abundant, this should be more accurately described in the text based on this data.

- The authors present convincing evidence showing that EV-bound VEGF is protected from bevacizumab. However, as detailed above, the relevance of this is unclear given the lower proportion of EV-bound VEGF compared to total secreted VEGF. Can the amount of bound/unbound drug be demonstrated for total conditioned media or plasma as for sEVs alone (fig 5d). Likewise, can the functional studies in 6e and f also be performed with conditioned media or plasma to assess the impact of this inhibition of endothelial cell phenotype.
- In Fig 6f, please provide representative images of tube formation.

Appropriateness and validity of statistical analysis:

- Overall, sufficient detail is provided in the figure legends regarding experimental replicates and stats for graphs.
- Since both parametric (t-test) and non-parametric (Mann Whitney test) pairwise comparisons are utilized, the authors should indicate whether and how they tested the data distribution to determine which test to use. This information would be fine in the Methods.
- Also, in many cases where more than two groups are being analyzed, parametric or non-parametric multiple comparison tests should be used.

Reviewer #2 (Remarks to the Author):

It is so enjoyable to read this excellent work. The study was well designed, experiments were carefully executed and the data support their conclusions.

The authors carefully designed each experiment to mechanistically explain their observations. They sequentially describe that small extracellular vesicles (sEVs) stimulates endothelial cell activity; define VEGF responsible for angiogenic activity; define VEGF189 as the particular isoform; clarify the specific sorting of VEGF189 to sEVs through heparin sulfate proteoglycans, show the sEVs surface VEGF189 resistant to bevacizumab, and finally clinical data of their findings. There is a complete set of data that are high clinically relevant. Another new mechanism of bevacizumab resistance.

I have only a couple of comments that may help the authors to continue their work along this line of thinking.

1) Why would tumor cells produce a sEVs surface VEGF to achieve their angiogenic functions? VEGF189 on tumor cell surface in interaction with tumor vessel endothelial cells would equally do this job. Would it be more reasonable and important for their distal functions, such as modulation of metastatic tumor growth in a remote organ? It would make a great sense to study this in a metastatic setting. Perhaps they should keep in mind for their future studies.

2) Would their work imply that ramucizumab is more effective than bevacizumab? A bevacizumab resistant cancer should switch to ramucizumab or other TKIs? Measurement of sEVs could serve as a marker for drug selection.

3) Among all growth factors, FGF-2 is the most potent heparin-binding factor with the highest heparin-binding affinity. According to their analysis, FGF-2 is also highly packed in sEVs. Surprisingly, FGF-2 is not on the surface of sEVs. Why? if the heparin sulfate proteoglycans is the general mechanism, FGF-2 should be on cell surface as well. FGF-2 is also a potent angiogenic factor that could contribute to resistance of bevacizumab. Perhaps the authors should further explain this. The present explanation is not convincing enough for readers.

4) If heparin and heparin sulfate are the neutralizing agents for bevacizumab, injection of heparin would cause resistance to bevacizumab treatment. This can be easily tested in a bevacizumab sensitive tumor model.

5) The authors should cite the following references related to antiangiogenic therapy and drug resistance (novel resistance mechanism not mentioned):

Cell Metab. 2018 Jun 5;27(6):1163-1165. doi: 10.1016/j.cmet.2018.05.014.

PMID: 29874563

Similar articles

Cell Metab. 2018 Jul 3;28(1):104-117.e5. doi: 10.1016/j.cmet.2018.05.005. Epub 2018 May 31.

Proc Natl Acad Sci U S A. 2017 Jun 27;114(26):E5226-E5235. doi: 10.1073/pnas.1705066114. Epub 2017 Jun 12.

Nat Commun. 2016 Sep 1;7:12680. doi: 10.1038/ncomms12680.

Chin J Cancer. 2016 Feb 15;35:21. doi: 10.1186/s40880-016-0084-4.

Nat Rev Endocrinol. 2014 Sep;10(9):530-9. doi: 10.1038/nrendo.2014.114. Epub 2014 Jul 22.

Proc Natl Acad Sci U S A. 2013 Jul 16;110(29):12018-23.

Sci Transl Med. 2011 Dec 21;3(114):114rv3. doi: 10.1126/scitranslmed.3003149.

Reviewer #3 (Remarks to the Author):

This study, if correctly performed, may be very interesting for the readership of this journal and it may have a high impact. It describes a mechanism of extracellular vesicle-mediated angiogenesis, a topic that is currently under-investigated.

Overall, the study lacks a solid prove that VEGF is associated with sEVs. The Suppl Fig 6 is crucial to prove the main point of the manuscript; but it lacks biochemical analysis; similarly the flow results are superficially described. The failure to show this set of data weakens the main message of the study. In addition, the immunofluorescence pictures are low quality.

Overall, the data do not support the conclusions.

The manuscript is superficially written and includes details that are inconsistent: is there a well-characterized CRISPR- cell line used as described in the Methods section? Are there GFP-cells used as described in the Methods section? And so on.. REVIEWERS' COMMENTS:

Reviewer #1 (Remarks to the Author):

The authors have satisfied my comments. I thank them for their thoroughness in addressing all issues.

Reviewer #2 (Remarks to the Author):

The authors have satisfactorily addressed my previous concerns. Congratulations on this wonderful contribution.

Reviewer #3 (Remarks to the Author):

The Authors have satisfactorily addressed all comments.

Cancer-derived small extracellular vesicles deliver heparin-bound, bevacizumab-insensitive VEGF independently of uptake

We sincerely thank the reviewers for their constructive comments and have extensively revised our manuscript to address all of the comments as described below. Major changes in the manuscript text are underlined. Please note that 'sEV-VEGF' refers to small extracellular vesicle-associated VEGF.

REVIEWER 1

In this manuscript, Ko et. al characterize the association of VEGF with sEVs and test the functional importance of this in cancer. They present interesting data showing a VEGF specific isoform association with sEVs that impacts VEGF stability and resistance to the drug bevacizumab. Overall, the experiments are clear and the data is well-presented. Additionally, this description of VEGF and sEVs adds to our knowledge of selective protein association with EVs and offers some new insight into distinctions between EV-associated and soluble proteins. Because VEGF and EVs are both areas of intense study in cancer, these findings are likely to be of interest to a wide audience. That said, there are a couple of major conceptual issues the authors should address experimentally or offer some discussion to better place their findings into proper context.

[1] *Although the authors do a nice job of validating their sEV preparations from cell lines following centrifugation, filtration, and density gradient ultracentrifugation in Fig 1 and Suppl Fig 1 and 2, some issues remain regarding their EV isolation and characterization. First, for human samples and experiments with PKH26 labeling, they have isolated sEVs using the Exoquick kit without showing any validation of those particular preparations. Because of the ability of Exoquick to purify non-EV impurities due to its polymer-based precipitation, Exoquick is not a commonly accepted method for sEV purification. Thus, it will be necessary for the authors to show better characterization of these particular sEV preparations, confirming they lack non-EV contaminants, or repeat these experiments using more conventional approaches i.e. ultracentrifugation.*

Exoquick reagent was only used to isolate sEVs from plasma samples of bevacizumab-treated patients with renal cell carcinoma **[Fig. 8g]**. In all other experiments including PKH26 labeling, sEVs were isolated from human clinical specimens, mouse samples and conditioned media by sequential filtration and density gradient ultracentrifugation. Volumes of plasma samples of bevacizumab-treated patients (which were from a clinical trial) were extremely limited and too small to be isolated by density gradient ultracentrifugation. We validated residual sEVs that were isolated from these plasma samples using Exoquick by Western blot and nanoparticle tracking analysis. Our data shows that these sEVs **(1)** express sEV markers, **(2)** lack markers of larger EVs and non-EV components **[Supp Fig. 13a]** and **(3)** have a particle size distribution that is highly similar to that of sEVs isolated by density gradient ultracentrifugation **[compare Supp Fig. 13b with 1b]**.

Second, although the authors confirm that VEGF is found in EV fractions using density gradient centrifugation, they describe this as being a marked enrichment in these fractions, when in fact VEGF is also equally abundant in other fractions (Suppl Fig 6 purple and gray bars).

We performed additional characterization of the density fractions. VEGF was detected in the highest density fractions that largely consisted of unfractionated material and soluble proteins **[Supp Fig. 8a]**. These fractions contained VEGF₁₂₁ and VEGF₁₆₅ but not VEGF₁₈₉ **[Supp Fig. 8b]**. Of the other fractions, only the fractions of the density of sEVs showed prominent levels of VEGF and this VEGF exclusively comprised of VEGF₁₈₉ **[Supp Fig. 8a-b]**.

Similarly, the authors also show that there is abundant non-sEV associated VEGF; therefore, effort should be invested into verifying that VEGF is actively sorted into EVs and is not sticking to them as a result of being co-secreted. While the authors provide evidence that VEGF is sticking to the surface of sEVs via heparin binding, this could be due to active packaging of VEGF into sEVs or, alternatively, binding of VEGF to heparin-containing sEVs in conditioned medium or circulation following secretion. This post-secretion interaction would still allow for EV-associated VEGF to be identified in EV fractions by density gradient separation. Furthermore, this interaction would be affected by perturbations that alter levels of heparin on sEV surfaces, so the experiments in Fig 5 are not necessarily specific to EV-sorted VEGF. Regardless of the route by which VEGF becomes associated with sEVs, this sEV-bound VEGF may still be important, as the authors go on to assess, but this distinction should be clarified or these possibilities discussed.

We agree that the association of VEGF with sEVs could occur via sorting into sEVs and/or binding to sEVs post-secretion. To test whether VEGF₁₈₉ (the predominant isoform of VEGF in sEVs) binds to sEVs following secretion as opposed to being sorted into sEVs, we incubated conditioned media of VEGF^{-/-} cancer cells (that contains secreted EVs but no VEGF) with recombinant VEGF₁₈₉ (i.e. 'free VEGF₁₈₉') that was added at an amount equivalent to the total amount of VEGF secreted by VEGF^{-/-} cancer cells that were stably transfected with VEGF₁₈₉. Thereafter, sEVs were isolated from the media. The amount of VEGF₁₈₉ detected in these sEVs (i.e. through post-secretion binding) was ~25% of the amount of VEGF₁₈₉ in sEVs secreted by VEGF₁₈₉-transfected VEGF^{-/-} cells **[Fig. 6d]**. We confirmed this finding using two different sets of cancer cell line models (ES2, ovarian; HCT116, colorectal). This new data suggests that, although VEGF₁₈₉ is able to bind to sEVs post-secretion, the presence of this ligand in sEVs predominantly occurs through selective sorting into these vesicles.

[2] *It is interesting that VEGF bound to heparin on sEVs is particularly stable and also resistant to bevacizumab. However, based on the data presented, the physiological relevance of this finding is unclear. First, although the authors show in Fig 5e that sEV-bound VEGF189 is more stable compared to free, recombinant VEGF189, this particular experiment does not accurately model a physiological situation in which it would be expected that VEGF or sEVs carrying VEGF are secreted on a more continuous basis. Thus, any VEGF that is turned over or degraded would be actively replaced, bringing into question the need for enhanced stability of VEGF via sEV secretion. Moreover, the authors have calculated in Table I that the majority of VEGF is not sEV-associated, suggesting that there is little necessity for this enhanced stability of VEGF through sEV secretion since most of the secreted VEGF is soluble. Along these lines, while VEGF189 was found to be particularly sEV-associated compared to other isoforms, the other isoforms are still secreted in soluble form (more so than 189 by HCT116 cells as shown in 5a). Finally, the authors suggest VEGF secreted with sEVs signals locally to endothelial cells, so again, the need for and impact of enhanced stability is unclear given the short-range paracrine signaling within the tumor microenvironment. Overall, it seems a small amount of VEGF would be protected by being bound to the surface of sEVs so the relevance of this protective effect, its relative contribution to the tumor microenvironment, and how it may promote bevacizumab resistance in a meaningful way is unclear.*

We understand the limitations of studies in which sEV-VEGF is exogenously added. To evaluate the relative contribution of sEV-VEGF to the effects of total VEGF, we performed *in vitro* studies that model conditions in which endogenous soluble VEGF and sEVs carrying VEGF are co-secreted by cancer cells, and also performed new *in vivo* studies:

In new *in vitro* studies, we used whole cancer cell-conditioned media (that contains all soluble factors and EVs that are secreted by cancer cells) and cancer cell-conditioned media that was depleted of sEVs. Analysis of VEGF levels in whole and sEV-depleted conditioned media revealed that sEV-VEGF constitutes ~35% of the total VEGF secreted by cancer cells **[Fig. 3f]**. This abundance of sEV-VEGF is within the range detected in body fluids of tumor-bearing mice and cancer patients **[Table I]**. By comparing the effects of stimulating endothelial cells with whole *versus* sEV-depleted conditioned media, we found that depletion of sEVs reduced VEGFR2 phosphorylation **[Fig. 3g]** and decreased tube formation by 30% **[Fig. 3h]**. This % decrease in endothelial cell responses is consistent with the % of total VEGF that comprises of sEV-VEGF.

We also tested the neutralizing ability of bevacizumab under conditions where endogenous soluble VEGF and sEVs carrying VEGF are co-secreted by cancer cells. Bevacizumab was incubated with whole or sEV-depleted cancer cell-conditioned media and unbound bevacizumab assayed thereafter. Levels of unbound bevacizumab in whole media were almost identical to those in sEV-depleted media **[Supp Fig. 11b]**, indicating that bevacizumab only neutralized non-sEV-VEGF. This was confirmed by the decrease in unbound bevacizumab levels following incubation in sEV-depleted media to which recombinant VEGF (i.e. 'non-sEV-VEGF') had been added at an amount equivalent to that of sEV-VEGF in whole media **[Supp Fig. 11b]**. Consistent with the ability of bevacizumab to bind non-sEV-VEGF and not sEV-VEGF, bevacizumab completely inhibited VEGFR2 phosphorylation in endothelial cells that were stimulated with sEV-depleted media but only partially inhibited VEGFR2 phosphorylation in endothelial cells that were stimulated with whole media **[Supp Fig. 11c]**.

Furthermore, we performed new studies using two different xenograft models to evaluate the effect of bevacizumab on endogenous levels of sEV-VEGF and non-sEV-VEGF that are co-secreted by tumors. We generated s.c. tumors in mice from parental ES2 (ovarian) and HCT116 (colorectal) cells and then treated mice with bevacizumab for one week. Bevacizumab did not inhibit tumor growth during this period **[Supp Fig. 12a]**, as likewise observed in other studies that used the same models (Tsujioka *et al* (2011) Anticancer Res; Iwai *et al* (2016) Oncol Rep). Samples of plasma, collected from mice pre-

and post-treatment, were assayed for human (i.e. tumor-derived) VEGF. Following bevacizumab treatment, levels of circulating non-sEV-VEGF significantly decreased, whereas levels of sEV-VEGF significantly increased [Supp Fig. 12c-d].

We would like to clarify that we did not suggest that sEV-VEGF only acts locally. Indeed, our findings that sEV-VEGF is present in the peripheral circulation of tumor-bearing mice [Supp Fig. 12d] and of cancer patients [Figs. 5d,8g] implicate sEV-VEGF in long-range signaling. This point is discussed in the Discussion.

Indeed, treatment of tumor-bearing mice in the absence of exogenously administered exosomes with the drug is sufficient to limit tumor growth and tumor angiogenesis (Fig 7), suggesting this protection of VEGF by endogenously secreted EVs is not enough to confer bevacizumab resistance.

In Fig. 8a-e (previously Fig. 7), xenografts were generated from VEGF^{-/-} human tumor cells. This model produces no tumor-derived VEGF (either soluble or EV-associated). In Groups 3 and 4, the sole source of human VEGF was exogenously administered recombinant VEGF (i.e. 'free VEGF'). In Groups 1 and 2, the sole source of human VEGF was carried by exogenously administered sEVs. The result that the reviewer refers to is where bevacizumab inhibited tumor growth that was stimulated by free VEGF (compare group 4 (bevacizumab-treated) with Group 3 (control Ig-treated)). In contrast, bevacizumab did not inhibit tumor growth that was stimulated by sEVs carrying VEGF (compare group 2 (bevacizumab-treated) with Group 1 (control Ig-treated)). As noted above, we performed additional studies using xenograft models that endogenously express VEGF to evaluate the significance of sEV-VEGF in bevacizumab resistance [Supp Fig. 12].

Although the patient data in 7d seems supportive of the author's conclusions, the majority of patients with progressing disease have sEV-VEGF levels comparable to patients with stable or regressing disease.

Baseline levels of total VEGF were not significantly different between patients with progressing disease and those who had stable or regressing disease following bevacizumab treatment [Fig. 8f, previously 7d]. In contrast, baseline levels of sEV-VEGF were significantly higher in patients with progressing disease than in those with stable or regressing disease ($P = 0.010$) [Fig. 8g].

Specific comments:

[3] *In the introduction the authors state that EV classification is based on size. However, it is also based on the cellular origin of the vesicles. Although the authors conservatively, and appropriately, use the term sEV to describe the vesicles they are studying since they have distinguished exosomes versus small microvesicles, it would be a necessary for them to clarify for their readers that classification is not just based on size, if they indeed want to make this statement.*

The authors should also consider reframing the goal of their study, as there is already published work on how VEGF carried by EVs affects endothelial cells (e.g., see Zhao et. al Biochem Biophys Res Commun 2018, Maisto et. al Cell Cycle 2019). Thus, the impact of this present work will not be to understand how sEVs affect "stromal cells through mechanisms other than transferring their luminal cargo..." as stated at the end of the introduction. This will provide readers with a broader picture of the field and references to other papers of interest.

In the Introduction, we have described the biogenesis and classification of EVs. We also clarified the starting point of our study. This was to investigate whether sEVs mediate intercellular communication in the tumor microenvironment via mechanisms other than transferring their luminal cargo (rather than investigating at the outset what VEGF does in sEVs). In the Discussion, we discuss recent reports that implicate VEGF and other proteins in the angiogenic activity of EVs.

[4] *The detailed characterization of the EVs the authors are studying is appreciated. In Fig 1b EM images, the authors should also provide a lower power or wider field image to give a sense for the proportion of vesicles that stain for CD63.*

At lower power or wider field, only ~10 EVs can be visualized at sufficient resolution. To evaluate the proportion of sEVs that are CD63-positive in a quantitative and unbiased manner, we directly stained sEVs with fluorochrome-conjugated CD63 Ab and analyzed staining in the gated population of sEVs by high-resolution flow cytometry. A minimum of 10,000 gated sEVs were analyzed per assay [Supp. Fig. 6b]. Using this approach, CD63 was detected on the surface of ~90% of sEVs secreted by two different types of cancer cells [Supp. Fig. 6c-d].

[5] *In Fig 1, it is surprising the inhibition of endocytosis does not affect endothelial cell migration and tube formation, as endocytosis of receptors that regulate motility is usually key for proper function and signaling of these receptors during migratory processes. Although the authors mention in the discussion that recycling/trafficking/endocytosis of VEGFR may not be necessary in this context, other membrane receptors involved in motility do, so it would be expected that these would be affected by the inhibitors. Can the authors explain this?*

Endocytosis inhibitors are commonly used to block uptake of sEVs but, as noted by the reviewer, can impair cell motility. In preliminary studies, we tested endocytosis inhibitors at various doses and incubation times to identify conditions where these inhibitors could inhibit sEV uptake without significantly inhibiting endothelial cell migration and tube formation. Cell migration and tube formation were not significantly inhibited by short-term treatment (4 to 5 hours) with inhibitors at the indicated doses but were inhibited when treatment was extended beyond 5 hours. We clarified this point in the Results.

[6] *Fluorescence microscopy images of the cells should also be provided along with the flow cytometry data in Fig 1 g, h and Suppl Fig 3, 4 to show how the sEVs interact with HUVECs and the differences in uptake with the inhibitors.*

We have included representative fluorescence microscopy images that show **(1)** kinetics of sEV uptake by HUVEC [**Supp Fig. 2a**] and **(2)** blockade of sEV uptake by endocytosis inhibitors [**Supp Fig. 3a**] to accompany the flow cytometry data.

[7] *Based on Fig 2b, VEGF is described as being the most abundant angiogenic factor on sEV surfaces. However, when using immunoassays of any sort, it is not possible to comment on the relative abundance of different factors, as this will be dictated by the performance of the antibody being used and different antibodies need to be used for different proteins. This point is underscored by comparing 2b to 2a, where the intensity of VEGF is actually the lowest while it gives the strongest signal by ELISA in 2b; this disparate expression signal is also evident for other factors measured, such as IL8, which, compared to VEGF, clearly gives a stronger signal in a but gives lower pg/ml measurements in b. If mass spec proteomics is performed, then it is possible to compare the levels of different proteins.*

The antibody array in Fig. 2a does not provide a read-out of the relative abundance of the different factors as each signal depends on the affinity of an individual antibody for a given factor. An important purpose of the ELISAs in Fig. 2b was to assay **(i)** lysates of sEVs to determine the total amount of a given factor in sEVs and **(ii)** equivalent amounts of intact sEVs to determine the amount of the factor on the surface of sEVs. Using this approach, we could gain insight into how much of a given factor is encapsulated within sEVs versus present on the sEV surface. This would not be possible by mass spectrometry. We have clarified these points in the Results.

[8] *For Fig 2c, please provide validation of VEGF depletion by western in the cells and ELISA in the supernatant.*

We have validated VEGF deletion in cells by Western blot [**Supp Fig. 4a**], and supernatants by ELISA [**Supp Fig. 4b**].

[9] *For Fig 2d, as with EM in Fig 1, low power/wide field images are necessary to assess the prevalence of EVs with VEGF. Use of EVs from VEGF negative cells would be a good control for this EM too.*

We described limitations of EM in our response in **[4]**. To evaluate the prevalence of sEVs with VEGF in a quantitative and unbiased manner, we evaluated direct staining of sEVs with VEGF Ab in the gated population of sEVs by flow cytometry (minimum of 10,000 gated sEVs per assay). For rigor, we analyzed sEVs of 2 sets of isogenic VEGF^{+/+} and VEGF^{-/-} cancer cells and included dual staining with CD63 Ab as a control. CD63 was detected on ~90% of both VEGF^{+/+} and VEGF^{-/-} sEVs. VEGF was not detected on VEGF^{-/-} sEVs but was detected on ~80% of VEGF^{+/+} sEVs [**Supp Fig. 6c-d**].

[10] *For Fig 2 f and g, representative images of the quantification should be presented. For these experiments, as well as 2e, it would also be helpful to use EVs from the VEGF negative cells presented in 2c.*

Quantification of tube formation showing effects of TKIs and VEGFR2 Ab is now presented in Fig 3b and 3d and is accompanied by representative images [**Fig. 3c,e**]. Quantification of tube formation in response to sEVs of isogenic VEGF^{+/+} and VEGF^{-/-} cancer cells is now shown in Fig 4b and is accompanied by representative images [**Fig. 4c**].

[11] *For Fig 4a and 4b, the authors draw comparisons between the expression levels of the different isoforms in cells and the isoform levels between cells and EVs. They state that 165 and 121 are the predominant forms in cells, but based on the blot in 4a, all isoforms seem equally abundant, this should be more accurately described in the text based on this data.*

We have revised the description of isoform abundance in the Results.

[12] *The authors present convincing evidence showing that EV-bound VEGF is protected from bevacizumab. However, as detailed above, the relevance of this is unclear given the lower proportion of EV-bound VEGF compared to total secreted VEGF. Can the amount of bound/unbound drug be demonstrated for total conditioned media or plasma as for sEVs alone (fig 5d). Likewise, can the functional studies in 6e and f also be performed with conditioned media or plasma to assess the impact of this inhibition of endothelial cell phenotype.*

As noted in our response in **[2]**, we evaluated the neutralizing ability of bevacizumab under conditions where endogenous soluble VEGF and sEVs carrying VEGF are co-secreted by cancer cells by incubating bevacizumab with whole or sEV-depleted cancer cell-conditioned media and then assaying unbound bevacizumab. Levels of unbound bevacizumab in whole media were almost identical to those in sEV-depleted media **[Supp Fig. 11b]**, indicating that bevacizumab only neutralized non-sEV-VEGF. Consistent with the ability of bevacizumab to bind non-sEV-VEGF and not sEV-VEGF, bevacizumab completely inhibited VEGFR2 phosphorylation in endothelial cells that were stimulated with sEV-depleted media but only partially inhibited VEGFR2 phosphorylation in endothelial cells stimulated with whole media **[Supp Fig. 11c]**. Furthermore, our findings that bevacizumab does not neutralize sEV-VEGF is supported by our new xenograft studies in which we evaluated the effect of bevacizumab on endogenous levels of sEV-VEGF and non-sEV-VEGF that are co-secreted by tumors **[Supp Fig. 12c-d]**.

[13] *In Fig 6f, please provide representative images of tube formation.*

Quantification of tube formation showing effects of bevacizumab is now presented in Fig 7f and is accompanied by representative images **[Fig. 7g]**.

[14] *Appropriateness and validity of statistical analysis:*

- *Overall, sufficient detail is provided in the figure legends regarding experimental replicates and stats for graphs.*
- *Since both parametric (t-test) and non-parametric (Mann Whitney test) pairwise comparisons are utilized, the authors should indicate whether and how they tested the data distribution to determine which test to use. This information would be fine in the Methods.*
- *Also, in many cases where more than two groups are being analyzed, parametric or non-parametric multiple comparison tests should be used.*

Normality of data distribution in groups was assessed by Shapiro-Wilk test. Where indicated, significance of data in *in vitro* and *in vivo* assays was assessed by one-way or two-way ANOVA with Bonferroni's corrections for multiple comparisons, or by unpaired or paired two-tailed Student's *t*-test. Significance of data between patient groups was assessed by Mann-Whitney *U*-test as data in these groups was non-normally distributed. These points have been clarified in the Methods and figure legends.

REVIEWER 2

It is so enjoyable to read this excellent work. The study was well designed, experiments were carefully executed and the data support their conclusions.

The authors carefully designed each experiment to mechanistically explain their observations. They sequentially describe that small extracellular vesicles (sEVs) stimulates endothelial cell activity; define VEGF responsible for angiogenic activity; define VEGF189 as the particular isoform; clarify the specific sorting of VEGF189 to sEVs through heparin sulfate proteoglycans, show the sEVs surface VEGF189 resistant to bevacizumab, and finally clinical data of their findings. There is a complete set of data that are high clinically relevant. Another new mechanism of bevacizumab resistance.

I have only a couple of comments that may help the authors to continue their work along this line of thinking.

[1] *Why would tumor cells produce a sEVs surface VEGF to achieve their angiogenic functions? VEGF189 on tumor cell surface in interaction with tumor vessel endothelial cells would equally do this job. Would it be more reasonable and*

important for their distal functions, such as modulation of metastatic tumor growth in a remote organ? It would make a great sense to study this in a metastatic setting. Perhaps they should keep in mind for their future studies.

It has been thought that VEGF₁₈₉, by virtue of being membrane/matrix-bound, acts locally whereas VEGF₁₂₁, by virtue of being freely secreted, mediates long-range signaling. Our findings that sEV-VEGF predominantly comprises of VEGF₁₈₉, is signaling-competent, highly stable and present in the peripheral circulation of tumor-bearing mice and cancer patients collectively support the possibility that VEGF₁₈₉, through being conveyed on the surface of secreted sEVs, could also mediate long-range signaling. At the present time, it is not possible to quantify the relative contributions of sEV-VEGF *versus* soluble VEGF in the local tumor microenvironment and at distal sites. However, it is quite conceivable that additional long-range signaling, mediated by VEGF₁₈₉ carried by sEVs, might provide a strong advantage to tumors and particularly for metastasis. We thank the reviewer for the thought-provoking comments and have discussed these points.

[2] *Would their work imply that ramucizumab is more effective than bevacizumab? A bevacizumab resistant cancer should switch to ramucizumab or other TKIs? Measurement of sEVs could serve as a marker for drug selection.*

We found that tube formation stimulated by sEVs carrying VEGF can be inhibited by VEGFR TKIs **[Fig. 3b-c]** and by a neutralizing antibody to VEGFR2 **[Fig. 3d-e]**, but not by bevacizumab **[Fig. 7f-g]**. The VEGFR2 antibody that we used was not humanized but, like ramucirumab, targets the extracellular domain of VEGFR2 and blocks ligand binding. Our work cannot be interpreted to imply that ramucirumab or TKIs would be more effective than bevacizumab as there are several mechanisms of resistance to anti-angiogenic therapy that are not necessarily unique to bevacizumab. However, as noted by the reviewer, our findings raise the possibility that treatment with ramucirumab or VEGFR TKIs instead of bevacizumab might be beneficial for patients who have elevated levels of sEV-VEGF. We have discussed this point in the Discussion.

[3] *Among all growth factors, FGF-2 is the most potent heparin-binding factor with the highest heparin-binding affinity. According to their analysis, FGF-2 is also highly packed in sEVs. Surprisingly, FGF-2 is not on the surface of sEVs. Why? if the heparin sulfate proteoglycans is the general mechanism, FGF-2 should be on cell surface as well. FGF-2 is also a potent angiogenic factor that could contribute to resistance of bevacizumab. Perhaps the authors should further explain this. The present explanation is not convincing enough for readers.*

An excellent point. Our finding that most of the FGF-2 in sEVs is not on the surface might be explained by the non-classical secretion mechanism of FGF-2. Whereas many other growth factors contain a signal peptide and undergo ER-to-Golgi trafficking, FGF-2 lacks a signal peptide and is instead recruited to the inner leaflet of the plasma membrane. Once there, FGF-2 undergoes oligomerization that in turn causes formation of membrane pores, enabling FGF-2 to cross the plasma membrane (LaVenuta *et al* (2015) J Biol Chem). FGF-2 is also released from cells in ectosomes that bud from the plasma membrane and range from 100 to 1000 nm in diameter (Taverna *et al* (2003) J Biol Chem). In accordance with recent MISEV2018 guidelines, we defined sEVs as EVs that are <200 nm in diameter. It is therefore possible that our sEV preparations not only included exosomes but also small ectosomes. The recruitment of FGF-2 to the inner leaflet and formation of ectosomes through outward budding of the plasma membrane could explain why FGF-2 is encapsulated inside these EVs rather than being on the surface.

[4] *If heparin and heparin sulfate are the neutralizing agents for bevacizumab, injection of heparin would cause resistance to bevacizumab treatment. This can be easily tested in a bevacizumab sensitive tumor model.*

Several residues in the β -sheets of VEGF are critical for binding to bevacizumab (Muller *et al* (1998) Structure), and the β -sheet content of VEGF substantially decreases when VEGF interacts with HMW heparin (Wijelath *et al* (2010) J Cell Biochem). We found that engagement of VEGF₁₈₉ to HMW heparin substantially reduces its recognition by bevacizumab **[Fig. 7a]**. Collectively, these findings suggest that interaction with heparin changes the conformation of VEGF, rendering it less recognizable by bevacizumab. In the experiment suggested by the reviewer, heparin would only increase resistance to bevacizumab under circumstances where a significant proportion of VEGF₁₈₉ that is secreted/shed by cancer cells is not bound (via heparin) to sEVs. However, our new data shows that ~90% of the total VEGF₁₈₉ that is secreted/shed by cancer cells is already sEV-bound **[Fig. 6b]**. In this case, adding heparin would be unlikely to increase resistance to bevacizumab.

[5] *The authors should cite the following references related to antiangiogenic therapy and drug resistance (novel resistance mechanism not mentioned):*

Cell Metab. 2018 Jun 5;27(6):1163-1165. doi: 10.1016/j.cmet.2018.05.014. PMID: 29874563

Cell Metab. 2018 Jul 3;28(1):104-117.e5. doi: 10.1016/j.cmet.2018.05.005. Epub 2018 May 31.

Proc Natl Acad Sci U S A. 2017 Jun 27;114(26):E5226-E5235. doi: 10.1073/pnas.1705066114. Epub 2017 Jun 12.

Nat Commun. 2016 Sep 1;7:12680. doi: 10.1038/ncomms12680.

Chin J Cancer. 2016 Feb 15;35:21. doi: 10.1186/s40880-016-0084-4.

Nat Rev Endocrinol. 2014 Sep;10(9):530-9. doi: 10.1038/nrendo.2014.114. Epub 2014 Jul 22.

Proc Natl Acad Sci U S A. 2013 Jul 16;110(29):12018-23.

Sci Transl Med. 2011 Dec 21;3(114):114rv3. doi: 10.1126/scitranslmed.3003149.

We thank the reviewer for this additional information. Due to space limitations, we are not able to include all references, but have discussed and cited most of the original research articles listed above and others cited in the reviews listed above.

REVIEWER #3

This study, if correctly performed, may be very interesting for the readership of this journal and it may have a high impact. It describes a mechanism of extracellular vesicle-mediated angiogenesis, a topic that is currently under-investigated.

[1] *Overall, the study lacks a solid prove that VEGF is associated with sEVs. The Suppl Fig 6 is crucial to prove the main point of the manuscript; but it lacks biochemical analysis; similarly the flow results are superficially described. The failure to show this set of data weakens the main message of the study. In addition, the immunofluorescence pictures are low quality. Overall, the data do not support the conclusions.*

As noted in our MS and by other reviewers, VEGF has been detected in sEVs by several groups (Skog *et al* (2008) *Nat Cell Biol*; Treps *et al* (2017) *J Extracell Vesicles*; Zhao *et al* (2018) *Biochem Biophys Res Commun*) and implicated in the angiogenic activity of sEVs. Our study addresses several critical gaps-in-knowledge: **(i)** what are the molecular characteristics of sEV-VEGF, **(ii)** do VEGF isoforms vary in their localization to sEVs, **(iii)** what are the mechanisms that regulate this localization, **(iv)** how does sEV-VEGF signal to recipient cells, **(v)** does association of VEGF with sEVs alter its biological activity, **(vi)** what is the significance of sEV-VEGF to anti-VEGF therapy. We have provided substantial new data in response to other reviewers' comments that strengthen our answers to these important biological and clinically relevant questions.

To eliminate the possibility that the presence of VEGF resulted from contamination during density gradient ultracentrifugation, we assayed all of the density fractions for VEGF and performed additional characterization. VEGF was detected in the highest density fractions that largely consisted of unfractionated material and soluble proteins [**Supp Fig. 8a**, previously Supp Fig. 6]. These fractions contained VEGF₁₂₁ and VEGF₁₆₅ but not VEGF₁₈₉ [**Supp Fig. 8b**]. Of the other fractions, only the fractions of the density of sEVs showed prominent levels of VEGF and this VEGF exclusively comprised of VEGF₁₈₉ [**Supp Fig. 8a-b**]. Localization of VEGF on the surface of sEVs is strongly supported by immunoassays [**Fig. 2b**] and flow cytometric studies [**Fig. 2c, Supp Fig. 5b-c**] that include multiple controls (VEGF-deficient sEVs, CD63 and TSG101 as positive and negative controls for sEV surface protein). We have also included compelling new data using a different approach in which direct staining of VEGF on the surface of sEVs was evaluated by high-resolution flow cytometry [**Supp Fig. 6c-d**]. Furthermore, our findings that localization of VEGF₁₈₉ to sEVs is diminished in cells deficient in heparin sulfate biosynthesis [**Fig. 6e**] and that heparinase removes VEGF from the surface of sEVs [**Fig. 6f-g**] strongly support the notion that VEGF₁₈₉ interacts with the surface of sEVs at least in part through heparin-binding. We have extensively revised the Results to describe our flow cytometry and other studies in more detail. Higher quality, wider field immunofluorescence images (i.e. of vessels in xenograft models) have been included [**Fig. 4e, 8b**].

The manuscript is superficially written and includes details that are inconsistent: is there a well-characterized CRISPR- cell line used as described in the Methods section? Are there GFP-cells used as described in the Methods section? And so on..

The generation and characterization of HCT116 VEGF^{-/-} cells is described in a previous report (Dang *et al* (2006) *Cancer Res*) which we cite in our MS. The generation of GFP-expressing ES2 VEGF^{-/-} cells is described in the Methods. We have validated VEGF deficiency in both of these cell lines [**Supp Fig. 4a-c**].

Again, we sincerely thank the reviewers and Editor for their constructive suggestions which have improved our study. We hope that we have satisfactorily addressed all of their comments.

REVIEWERS' COMMENTS:

Reviewer #1 (Remarks to the Author):

The authors have satisfied my comments. I thank them for their thoroughness in addressing all issues.

Reviewer #2 (Remarks to the Author):

The authors have satisfactorily addressed my previous concerns. Congratulations on this wonderful contribution.

Reviewer #3 (Remarks to the Author):

The Authors have satisfactorily addressed all comments.